# Sqrt(d) Dimension Dependence of Langevin Monte Carlo

**Ruilin Li**
Georgia Institute of Technology
ruilin.li@gatech.edu

**Hongyuan Zha**
School of Data Science
Shenzhen Institute of Artificial Intelligence and Robotics for Society
The Chinese University of Hong Kong, Shenzhen
zhahy@cuhk.edu.cn

**Molei Tao**
Georgia Institute of Technology
mtao@gatech.edu

## Abstract

This article considers the popular MCMC method of unadjusted Langevin Monte Carlo (LMC) and provides a non-asymptotic analysis of its sampling error in 2-Wasserstein distance. The proof is based on a refinement of mean-square analysis in Li et al. (2019), and this refined framework automates the analysis of a large class of sampling algorithms based on discretizations of contractive SDEs. Using this framework, we establish an $\widetilde{\mathcal{O}}\left(\sqrt{d}/\epsilon\right)$ mixing time bound for LMC, without warm start, under the common log-smooth and log-strongly-convex conditions, plus a growth condition on the 3rd-order derivative of the potential of target measures. This bound improves the best previously known $\widetilde{\mathcal{O}}\left(d/\epsilon\right)$ result and is optimal (in terms of order) in both dimension $d$ and accuracy tolerance $\epsilon$ for target measures satisfying the aforementioned assumptions. Our theoretical analysis is further validated by numerical experiments.

## 1 Introduction

The problem of sampling statistical distributions has attracted considerable attention, not only in the fields of statistics and scientific computing, but also in machine learning (Robert & Casella, 2013; Andrieu et al., 2003; Liu, 2008); for example, how various sampling algorithms scale with the dimension of the target distribution is a popular recent topic in statistical deep learning (see discussions below for references). For samplers that can be viewed as discretizations of SDEs, the idea is to use an ergodic SDE whose equilibrium distribution agrees with the target distribution, and employ an appropriate numerical algorithm that discretizes (the time of) the SDE. The iterates of the numerical algorithm will approximately follow the target distribution when converged, and can be used for various downstream applications such as Bayesian inference and inverse problem (Dashti & Stuart, 2017). One notable example is the Langevin Monte Carlo algorithm (LMC), which corresponds to an Euler-Maruyama discretization of the overdamped Langevin equation. Its study dated back to at least the 90s (Roberts et al., 1996) but keeps on leading to important discoveries, for example, on non-asymptotics and dimension dependence, which are relevant to machine learning (e.g., Dalalyan (2017a;b); Cheng et al. (2018a); Durmus et al. (2019); Durmus & Moulines (2019); Vempala & Wibisono (2019); Dalalyan & Riou-Durand (2020); Li et al. (2019); Erdogdu & Hosseinzadeh (2021); Mou et al. (2019); Lehec (2021)). LMC is closely related to SGD too (e.g., Mandt et al. (2017)). Many other examples exist, based on alternative SDEs and/or different discretizations (e.g., Dalalyan & Riou-Durand (2020); Ma et al. (2021); Mou et al. (2021); Li et al. (2020); Roberts & Rosenthal (1998); Chewi et al. (2021); Shen & Lee (2019)).

Quantitatively characterizing the non-asymptotic sampling error of numerical algorithms is usually critical for choosing the appropriate algorithm for a specific downstream application, for providing practical guidance on hyperparameter selection and experiment design, and for designing improved samplers. A powerful tool that dates back to (Jordan et al., 1998) is a paradigm of non-asymptotic

error analysis, namely to view sampling as optimization in probability space, and it led to many important recent results (e.g., Liu & Wang (2016); Dalalyan (2017a); Wibisono (2018); Zhang et al. (2018); Frogner & Poggio (2020); Chizat & Bach (2018); Chen et al. (2018); Ma et al. (2021); Erdogdu & Hosseinzadeh (2021)). It works by choosing an objective functional, typically some statistical distances/divergences, and showing that the law of the iterates of sampling algorithms converges in that objective functional. However, the choice of the objective functional often needs to be customized for different sampling algorithms. For example, KL divergence works for LMC (Cheng & Bartlett, 2018), but a carefully hand-crafted cross term needs to be added to KL divergence for analyzing KLMC (Ma et al., 2021). Even for the same underlying SDE, different discretization schemes exist and lead to different sampling algorithms, and the analyses of them had usually been case by case (e.g., Cheng et al. (2018b); Dalalyan & Riou-Durand (2020); Shen & Lee (2019)). Therefore, it would be a desirable complement to have a unified, general framework to study the non-asymptotic error of SDE-based sampling algorithms. Toward this goal, an alternative approach to analysis has recently started attracting attention, namely to resort to the numerical analysis of SDE integrators (e.g., Milstein & Tretyakov (2013); Kloeden & Platen (1992)) and quantitatively connect the integration error to the sampling error. One remarkable work in this direction is Li et al. (2019), which will be discussed in greater details later on.

The main tool of analysis in this paper will be a strengthened version (in specific aspects that will be clarified soon) of the result in Li et al. (2019). Although this analysis framework is rather general and applicable to a broad family of numerical methods that discretize contractive[1] SDEs, the main innovation focuses on a specific sampling algorithm, namely LMC, which is widely used in practice. Its stochastic gradient version is implemented in common machine learning systems, such as Tensorflow (Abadi et al., 2016), and is the off-the-shelf algorithm for large scale Bayesian inference. With the ever-growing size of parameter space, the non-asymptotic error of LMC is of central theoretical and practical interest, in particular, its dependence on the dimension of the sample space. The best current known upper bound of the mixing time in 2-Wasserstein distance for LMC is $\widetilde{\mathcal{O}}\left(\frac{d}{\epsilon}\right)$ (Durmus & Moulines, 2019). Motivated by a recent result (Chewi et al., 2021) that shows better dimension dependence for a Metropolis-Adjusted improvement of LMC, we will investigate if the current bound for (unadjusted) LMC is tight, and if not, what is the optimal dimension dependence.

**Our contributions**

The main contribution of this work is an improved $\widetilde{\mathcal{O}}\left(\frac{\sqrt{d}}{\epsilon}\right)$ mixing time upper bound for LMC in 2-Wasserstein distance, under reasonable regularity assumptions. More specifically, we study LMC for sampling from a Gibbs distribution $d\mu \propto \exp\left(-f(\boldsymbol{x})\right) d\boldsymbol{x}$. Under the standard smoothness and strong-convexity assumptions, plus an additional linear growth condition on the third-order derivative of the potential (which also shares connections to popular assumptions in the frontier literature), our bound improves upon the previously best known $\widetilde{\mathcal{O}}\left(\frac{d}{\epsilon}\right)$ result (Durmus & Moulines, 2019) in terms of dimension dependence. For a comparison, note it was known that discretized **kinetic** Langevin dynamics can lead to $\sqrt{d}$ dependence on dimension (Cheng & Bartlett, 2018; Dalalyan & Riou-Durand, 2020) and some believe that it is the introduction of momentum that improves the dimension dependence, but our result shows that discretized overdamped Langevin (no momentum) can also have mixing time scaling like $\sqrt{d}$. In fact, it is important to mention that recently shown was that Metropolis-Adjusted Euler-Maruyama discretization of **overdamped** Langevin (i.e., MALA) has an optimal dimension dependence of $\widetilde{\mathcal{O}}\left(\sqrt{d}\right)$ (Chewi et al., 2021), while what we analyze here is the **unadjusted** version (i.e., LMC), and it has the same dimension dependence (note however that our $\epsilon$ dependence is not as good as that for MALA; more discussion in Section 4). We also constructed an example which shows that the mixing time of LMC is at least $\widetilde{\Omega}\left(\frac{\sqrt{d}}{\epsilon}\right)$. Hence, our mixing time bound has the optimal dependence on both $d$ and $\epsilon$, in terms of order, for the family of target measures satisfying those regularity assumptions. Our theoretical analysis is further validated by empirical investigation of numerical examples.

A minor contribution of this work is the error analysis framework that we use. It is based on the classical mean-square analysis (Milstein & Tretyakov, 2013) in numerical SDE literature, however extended from finite time to infinite time. It is a minor contribution because this extension was

---

[1] possibly after a coordinate transformation

already pioneered in the milestone work of Li et al. (2019), although we will develop a refined version. Same as in classical mean-square analysis and in Li et al. (2019), the final (sampling in this case) error is only half order lower than the order of local strong integration error ($p_2$). This will lead to a $\widetilde{\mathcal{O}}\left(C^{\frac{1}{p_2-\frac{1}{2}}}/\epsilon^{\frac{1}{p_2-\frac{1}{2}}}\right)$ mixing time upper bound in 2-Wasserstein distance for the family of algorithms, where $C$ is a constant containing various information of the underlying problem, e.g., the dimension $d$. Nevertheless, the following two are **new** to this paper: (i) We weakened the requirement on local strong and weak errors. More precisely, Li et al. (2019) requires uniform bounds on local errors, but this could be a nontrivial requirement for SDE integrators; the improvement here only requires non-uniform bounds (although establishing the same result consequently needs notably more efforts, these are included in this paper too). (ii) The detailed expressions of our bounds are not the same as those in Li et al. (2019) (even if local errors could be uniformly bounded), and as we are interested in dimension-dependence of LMC, we work out constants and carefully track their dimension-dependences. Bounds and constants in Li et al. (2019) might not be specifically designed for tightly tracking dimension dependences, as the focus of their seminal paper was more on $\epsilon$ dependence; consequently, its general error bound only led to a $\tilde{O}(d)$-dependence in mixing time when applied to LMC (see Example 1 in Li et al. (2019)), whereas our result leads to $\tilde{O}(\sqrt{d})$.

## 2 PRELIMINARIES

**Notation**   Use symbol $\boldsymbol{x}$ to denote a $d$-dim. vector, and plain symbol $x$ to denote a scalar variable. $\|\boldsymbol{x}\|$ denotes the Euclidean vector norm. A numerical algorithm is denoted by $\mathcal{A}$ and its $k$-th iterate by $\bar{\boldsymbol{x}}_k$. Slightly abuse notation by identifying measures with their density function w.r.t. Lebesgue measure. Use the convention $\widetilde{\mathcal{O}}(\cdot) = \mathcal{O}(\cdot)\log^{\mathcal{O}(1)}(\cdot)$ and $\widetilde{\Omega}(\cdot) = \Omega(\cdot)\log^{\mathcal{O}(1)}(\cdot)$, i.e., the $\widetilde{\mathcal{O}}(\cdot)/\widetilde{\Omega}(\cdot)$ notation ignores the dependence on logarithmic factors. Use the notation $\widetilde{\Omega}(\cdot)$ similarly. Denote 2-Wasserstein distance by $W_2(\mu_1, \mu_2) = \left(\inf_{(\boldsymbol{X},\boldsymbol{Y})\sim\Pi(\mu_1,\mu_2)} \mathbb{E}\left\|\boldsymbol{X} - \boldsymbol{Y}\right\|^2\right)^{\frac{1}{2}}$, where $\Pi(\mu_1, \mu_2)$ is the set of couplings, i.e. all joint measures with $X$ and $Y$ marginals being $\mu_1$ and $\mu_2$. Denote the target distribution by $\mu$ and the law of a random variable $\boldsymbol{X}$ by $\mathrm{Law}(\boldsymbol{X})$. Finally, denote the mixing time of an sampling algorithm $\mathcal{A}$ converging to its target distribution $\mu$ in 2-Wasserstein distance by $\tau_{\mathrm{mix}}(\epsilon; W_2; \mathcal{A}) = \inf\{k \geq 0 | W_2(\mathrm{Law}(\bar{\boldsymbol{x}}_k), \mu) \leq \epsilon\}$.

**SDE for Sampling**   Consider a general SDE

$$d\boldsymbol{x}_t = \boldsymbol{b}(t, \boldsymbol{x}_t)dt + \boldsymbol{\sigma}(t, \boldsymbol{x}_t)d\boldsymbol{B}_t \tag{1}$$

where $\boldsymbol{b} \in \mathbb{R}^d$ is a drift term, $\boldsymbol{\sigma} \in \mathbb{R}^{d \times l}$ is a diffusion coefficient matrix and $\boldsymbol{B}_t$ is a $l$-dimensional Wiener process. Under mild condition (Pavliotis, 2014, Theorem 3.1), there exists a unique strong solution $\boldsymbol{x}_t$ to Eq. (1). Some SDEs admit geometric ergodicity, so that their solutions converge exponentially fast to a unique invariant distribution, and examples include the classical overdamped and kinetic Langevin dynamics, but are not limited to those (e.g., Mou et al. (2021); Li et al. (2020)). Such SDE are desired for sampling purposes, because one can set the target distribution to be the invariant distribution by choosing an SDE with an appropriate potential, and then solve the solution $\boldsymbol{x}_t$ of the SDE and push the time $t$ to infinity, so that (approximate) samples of the target distribution can be obtained. Except for a few known cases, however, explicit solutions of Eq. (1) are elusive and we have to resort to numerical schemes to simulate/integrate SDE. Such example schemes include, but are not limited to Euler-Maruyama method, Milstein methods and Runge-Kutta method (e.g., Kloeden & Platen (1992); Milstein & Tretyakov (2013)). With constant stepsize $h$ and at $k$-th iteration, a typical numerical algorithm takes a previous iterate $\bar{\boldsymbol{x}}_{k-1}$ and outputs a new iterate $\bar{\boldsymbol{x}}_k$ as an approximation of the solution $\boldsymbol{x}_t$ of Eq. (1) at time $t = kh$.

**Langevin Monte Carlo Algorithm**   LMC algorithm is defined by the following update rule

$$\bar{\boldsymbol{x}}_k = \bar{\boldsymbol{x}}_{k-1} - h\nabla f(\bar{\boldsymbol{x}}_{k-1}) + \sqrt{2h}\boldsymbol{\xi}_k, \quad k = 1, 2, \cdots \tag{2}$$

where $\{\boldsymbol{\xi}_k\}_{k\in\mathbb{Z}_{>0}}$ are i.i.d. standard $d$-dimensional Gaussian vectors. LMC corresponds to an Euler-Maruyama discretization of the continuous overdamped Langevin dynamics $d\boldsymbol{x}_t = -\nabla f(\boldsymbol{x}_t)dt + \sqrt{2}d\boldsymbol{B}_t$, which converges to an equilibrium distribution $\mu \sim \exp(-f(\boldsymbol{x}))$.

Dalalyan (2017b) provided a non-asymptotic analysis of LMC. An $\widetilde{\mathcal{O}}\left(\frac{d}{\epsilon^2}\right)$ mixing time bound in $W_2$ for log-smooth and log-strongly-convex target measures (Dalalyan, 2017a; Cheng et al., 2018a; Durmus et al., 2019) has been established. It was further improved to $\widetilde{\mathcal{O}}\left(\frac{d}{\epsilon}\right)$ under an additional Hessian Lipschitz condition (Durmus & Moulines, 2019). Mixing time bounds of LMC in other statistical distances/divergences have also been studied, including total variation distance (Dalalyan, 2017b; Durmus & Moulines, 2017) and KL divergence (Cheng & Bartlett, 2018).

**Classical Mean-Square Analysis**   A powerful framework for quantifying the *global* discretization error of a numerical algorithm for Eq. (1), i.e., $e_k = \left\{\mathbb{E}\left\|\boldsymbol{x}_{kh} - \bar{\boldsymbol{x}}_k\right\|\right\}^{\frac{1}{2}}$, is mean-square analysis (e.g., Milstein & Tretyakov (2013)). Mean-square analysis studies how *local* integration error propagate and accumulate into global integration error; in particular, if one-step (local) weak error and strong error (both the exact and numerical solutions start from the same initial value $\boldsymbol{x}$) satisfy

$$\|\mathbb{E}\boldsymbol{x}_h - \mathbb{E}\bar{\boldsymbol{x}}_1\| \leq C_1 \left(1 + \mathbb{E}\|\boldsymbol{x}\|^2\right)^{\frac{1}{2}} h^{p_1}, \quad \text{(local weak error)}$$

$$\left(\mathbb{E}\|\boldsymbol{x}_h - \bar{\boldsymbol{x}}_1\|^2\right)^{\frac{1}{2}} \leq C_2 \left(1 + \mathbb{E}\|\boldsymbol{x}\|^2\right)^{\frac{1}{2}} h^{p_2}, \quad \text{(local strong error)}$$

(3)

over a time interval $[0, Kh]$ for some constants $C_1, C_2 > 0$, $p_2 \geq \frac{1}{2}$ and $p_1 \geq p_2 + \frac{1}{2}$, then the global error is bounded by $e_k \leq C\left(1 + \mathbb{E}\|\boldsymbol{x}_0\|^2\right)^{\frac{1}{2}} h^{p_2 - \frac{1}{2}}$, $k = 1, \cdots, K$ for some constant $C > 0$ dependent on $Kh$.

Although classical mean-square analysis is only concerned with numerical integration error, sampling error can be also inferred. However, there is a limitation that prevents its direct employment in analyzing sampling algorithms: the global error bound only holds in finite time because the constant $C$ can grow exponentially as $K$ increases, rendering the bound useless when $K \to \infty$.

## 3   MEAN-SQUARE ANALYSIS OF SAMPLERS BASED ON CONTRACTIVE SDE

We now review and present some improved results on how to use mean-square analysis of **integration** error to quantify **sampling** error. A seminal paper in this direction is Li et al. (2019). What is known / new will be clarified. In all cases, the first step is to lift the finite time limitation when the SDE being discretized has some decaying property so that local integration errors do not amplify with time.

The specific type of decaying property we will work with is contractivity (after coordinate transformation). It is a sufficient condition for the underlying SDE to converge to a statistical distribution.

**Definition 3.1.** *A stochastic differential equation is contractive if there exists a non-singular constant matrix $A \in \mathbb{R}^{d \times d}$, a constant $\beta > 0$, such that any pair of solutions of the SDE satisfy*

$$\left(\mathbb{E}\left\|A\left(\boldsymbol{x}_t - \boldsymbol{y}_t\right)\right\|^2\right)^{\frac{1}{2}} \leq \left(\mathbb{E}\left\|A\left(\boldsymbol{x} - \boldsymbol{y}\right)\right\|^2\right)^{\frac{1}{2}} \exp(-\beta t),$$

(4)

*where $\boldsymbol{x}_t, \boldsymbol{y}_t$ are two solutions, driven by the same Brownian motion but evolved respectively from initial conditions $\boldsymbol{x}$ and $\boldsymbol{y}$.*

**Remark.** *As long as $\boldsymbol{b}$ and $\boldsymbol{\sigma}$ in (1) are not explicitly dependent on time, it suffices to find an arbitrarily small $t_0 > 0$ and show (4) holds for all $t < t_0$.*

**Remark.** *Sometimes contraction is not easy to establish directly, but can be shown after an appropriate coordinate transformation, see (Dalalyan & Riou-Durand, 2020, Proposition 1) for such a treatment for kinetic Langevin dynamics. The introduction of $A$ permits such transformations.*

In particular, overdamped Langevin dynamics, of which LMC is a discretization, is contractive when $f$ is strongly convex and smooth.

We now use contractivity to remove the finite time limitation. We first need a short time lemma.

**Lemma 3.2.** *(Milstein & Tretyakov, 2013, Lemma 1.3) Suppose $\boldsymbol{b}$ and $\boldsymbol{\sigma}$ in Eq.(1) are Lipschitz continuous. For two solutions $\boldsymbol{x}_t, \boldsymbol{y}_t$ of Eq. (1) starting from $\boldsymbol{x}, \boldsymbol{y}$ respectively, denote $\boldsymbol{z}_t(\boldsymbol{x}, \boldsymbol{y}) := (\boldsymbol{x}_t - \boldsymbol{x}) - (\boldsymbol{y}_t - \boldsymbol{y})$, then there exist $C_0 > 0$ and $h_0 > 0$ such that*

$$\mathbb{E}\left\|\boldsymbol{z}_t(\boldsymbol{x}, \boldsymbol{y})\right\|^2 \leq C_0 \left\|\boldsymbol{x} - \boldsymbol{y}\right\|^2 t, \quad \forall \boldsymbol{x}, \boldsymbol{y}, \ 0 < t \leq h_0.$$

(5)

Then we have a sequence of results that connects statistical property with integration property. We will see that a non-asymptotic **sampling** error analysis only requires bounding the orders of local weak and strong **integration** errors (if the continuous dynamics can be shown contractive).

**Theorem 3.3.** *(**Global Integration Error, Infinite Time Version**) Suppose Eq.(1) is contractive with rate $\beta$ and with respect to a non-singular matrix $A \in \mathbb{R}^{d \times d}$, with Lipschitz continuous $\boldsymbol{b}$ and $\boldsymbol{\sigma}$, and there is a numerical algorithm $\mathcal{A}$ with step size $h$ simulating the solution $\boldsymbol{x}_t$ of the SDE, whose iterates are denoted by $\bar{\boldsymbol{x}}_k, k = 0, 1, \cdots$. Suppose there exists $0 < h_0 \leq 1, C_1, C_2 > 0, D_1, D_2 \geq 0, p_1 \geq 1, \frac{1}{2} < p_2 \leq p_1 - \frac{1}{2}$ such that for any $0 < h \leq h_0$, the algorithm $\mathcal{A}$ has, respectively, local weak and strong error of order $p_1$ and $p_2$, defined as*

$$\begin{cases} \left\| \mathbb{E} \left( \boldsymbol{x}_h - \bar{\boldsymbol{x}}_1 \right) \right\| \leq \left( C_1 + D_1 \sqrt{\mathbb{E} \left\| \boldsymbol{x} \right\|^2} \right) h^{p_1}, \\ \left( \mathbb{E} \left\| \boldsymbol{x}_h - \bar{\boldsymbol{x}}_1 \right\|^2 \right)^{\frac{1}{2}} \leq \left( C_2^2 + D_2^2 \mathbb{E} \left\| \boldsymbol{x} \right\|^2 \right)^{\frac{1}{2}} h^{p_2}, \end{cases} \tag{6}$$

*where $\boldsymbol{x}_h$ solves Eq.(1) with any initial value $\boldsymbol{x}$ and $\bar{\boldsymbol{x}}_1$ is the result of applying $\mathcal{A}$ to $\boldsymbol{x}$ for one step.*

*If the solution of SDE $\boldsymbol{x}_t$ and algorithm $\mathcal{A}$ both start from $\boldsymbol{x}_0$, then for $0 < h \leq h_1 \triangleq$*

$$\min \left\{ h_0, \frac{1}{4\beta}, \left( \frac{\sqrt{\beta}}{4\sqrt{2}\kappa_A D_2} \right)^{\frac{1}{p_2 - \frac{1}{2}}}, \left( \frac{\beta}{8\sqrt{2}\kappa_A^2 (D_1 + C_0 D_2)} \right)^{\frac{1}{p_2 - \frac{1}{2}}} \right\}, \text{ the global error } \boldsymbol{e}_k \text{ is bounded as}$$

$$e_k := \left( \mathbb{E} \| \boldsymbol{x}_{kh} - \bar{\boldsymbol{x}}_k \|^2 \right)^{\frac{1}{2}} \leq C h^{p_2 - \frac{1}{2}}, \quad k = 0, 1, 2, \cdots, \qquad \text{where} \tag{7}$$

$$C = \frac{2}{\sqrt{\beta}} \kappa_A^2 \left( \frac{C_1 + C_0 C_2 + \sqrt{2} U (D_1 + C_0 D_2)}{\sqrt{\beta}} + C_2 + \sqrt{2} D_2 U \right), \tag{8}$$

*$C_0$ is from Eq. (5), $\kappa_A$ is the condition number of matrix $A$ and $U^2 \triangleq 4\mathbb{E} \| \boldsymbol{x}_0 \|^2 + 6 \mathbb{E}_\mu \| \boldsymbol{x} \|^2$.*

**Remark** (what's new)**.** *Thm.3.3 refines the seminal results in Li et al. (2019) in the sense that it only requires non-uniform bounds on the local error (6), whereas Li et al. (2019) requires uniform bounds, i.e., $D_1 = D_2 = 0$ in (6). Therefore, the refinement we present has wider applicability.*

*In general, local errors tend to depend on the current step's value, i.e. $D_1 \neq 0, D_2 \neq 0$. Allowing local error bounds to be non-uniform enabled applications such as proving the vanishing bias of mirror Langevin algorithm (Li et al., 2021). For a simpler illustration, consider LMC for 1D standard Gaussian target distribution, then we have $\left\| \mathbb{E} \left( \boldsymbol{x}_h - \bar{\boldsymbol{x}}_1 \right) \right\| = (e^{-h} - 1 + h) \mathbb{E} \| \boldsymbol{x} \| = (\frac{h^2}{2} + o(h^2)) \mathbb{E} \| \boldsymbol{x} \|$. One can see that the local error does depend on $\boldsymbol{x}$ and is not uniform. Meanwhile, our non-uniform condition still holds because $\left( \frac{h^2}{2} + o(h^2) \right) \mathbb{E} \| \boldsymbol{x} \| \leq (\frac{h^2}{2} + o(h^2)) \sqrt{\mathbb{E} \| \boldsymbol{x} \|^2}$ (and thus $p_1 = 2$). Note if the discretization does converge to a neighborhood of the target distribution, it is possible that $\mathbb{E} \| \boldsymbol{x} \|^2$ and/or $\mathbb{E} \| \boldsymbol{x} \|$ become bounded near the convergence, and in this case the $D_1$, $D_2$ parts can be absorbed into $C_1$ and $C_2$; however, this 'if' clause is exactly what we'd like to prove.*

*Nevertheless, we state for rigor that the convention $1/0 = \infty$ is used when $D_1 = D_2 = 0$. Another remark is, even in this case, our bound has a different expression from the seminal results. We will carefully work out, track, and combine dimension-dependence of constants using our bound.*

Following Theorem 3.3, we obtain the following non-asymptotic bound of the sampling error in $W_2$:

**Theorem 3.4.** *(**Non-Asymptotic Sampling Error Bound: General Case**) Under the same assumption and with the same notation of Theorem 3.3, we have*

$$W_2(Law(\bar{\boldsymbol{x}}_k), \mu) \leq e^{-\beta kh} W_2(Law(\boldsymbol{x}_0), \mu) + C h^{p_2 - \frac{1}{2}}, \quad \forall 0 < h \leq h_1.$$

A corollary of Theorem 3.4 is a bound on the mixing time of the sampling algorithm:

**Corollary 3.5.** *(**Upper Bound of Mixing Time: General Case**) Under the same assumption and with the same notation of Theorem 3.3, we have*

$$\tau_{\mathrm{mix}}(\epsilon; W_2; \mathcal{A}) \leq \max \left\{ \frac{1}{\beta h_1}, \frac{1}{\beta} \left( \frac{2C}{\epsilon} \right)^{\frac{1}{p_2 - \frac{1}{2}}} \right\} \log \frac{2W_2(Law(\boldsymbol{x}_0), \mu)}{\epsilon}$$

*In particular, when high accuracy is needed, i.e., $\epsilon < 2Ch_1^{p_2-\frac{1}{2}}$, we have*

$$\tau_{\mathrm{mix}}(\epsilon; W_2; \mathcal{A}) \leq \frac{(2C)^{\frac{1}{p_2-\frac{1}{2}}}}{\beta} \frac{1}{\epsilon^{\frac{1}{p_2-\frac{1}{2}}}} \log \frac{2W_2(Law(\boldsymbol{x}_0), \mu)}{\epsilon} = \widetilde{\mathcal{O}}\left(\frac{C^{\frac{1}{p_2-\frac{1}{2}}}}{\beta} \frac{1}{\epsilon^{\frac{1}{p_2-\frac{1}{2}}}}\right). \quad (9)$$

Corollary 3.5 states how mixing time depends on the order of local (strong) error (i.e., $p_2$) of a numerical algorithm. The larger $p_2$ is, the shorter the mixing time of the algorithm is, in term of the dependence on accuracy tolerance parameter $\epsilon$. It is important to note that for constant stepsize discretizations that are deterministic on the filtration of the driving Brownian motion and use only its increments, there is a strong order barrier, namely $p_2 \leq 1.5$ (Clark & Cameron, 1980; Rüemelin, 1982); however, methods involving multiple stochastic integrals (e.g., Kloeden & Platen (1992); Milstein & Tretyakov (2013); Rößler (2010)) can yield a larger $p_2$, and randomization (e.g., Shen & Lee (2019)) can possibly break the barrier too.

The constant $C$ defined in Eq. (7) typically contains rich information about the underlying SDE, e.g. dimension, Lipschitz constant of drift and noise diffusion, and the initial value $\boldsymbol{x}_0$ of the sampling algorithm. Through $C$, we can uncover the dependence of mixing time bound on various parameters, such as the dimension $d$. This will be detailed for Langevin Monte Carlo in the next section.

It is worth clarifying that once Thm.3.3 is proved, establishing Theorem 3.4 and Corollary 3.5 is relatively easy. In fact, analogous results have already been provided in Li et al. (2019), although they also required uniform local errors as consequences of their Thm.1. Nevertheless, we do not claim novelty in Theorem 3.4 and Corollary 3.5 and they are just presented for completeness. Our main refinement is just Thm.3.3 over Thm.1 in Li et al. (2019), and the non-triviality lies in its proof.

## 4 Non-Asymptotic Analysis of Langevin Monte Carlo Algorithm

We now quantify how LMC samples from Gibbs target distribution $\mu \sim \exp(-f(\boldsymbol{x}))$ that has a finite second moment, i.e., $\int_{\mathbb{R}^d} \|\boldsymbol{x}\|^2 \, d\mu < \infty$. Assume without loss of generality that the origin is a local minimizer of $f$, i.e. $\nabla f(\boldsymbol{0}) = \boldsymbol{0}$; this is for notational convenience in the analysis and can be realized via a simple coordinate shift, and it is not needed in the practical implementation. In addition, we assume the following two conditions hold:

**A 1.** (***Smoothness and Strong Convexity***) *Assume $f \in \mathcal{C}^2$ and is $L$-smooth and $m$-strongly-convex, i.e. there exists $0 < m \leq L$ such that $mI_d \preccurlyeq \nabla^2 f(\boldsymbol{x}) \preccurlyeq LI_d, \quad \forall \boldsymbol{x} \in \mathbb{R}^d$.*

Denote the condition number of $f$ by $\kappa \triangleq \frac{L}{m}$. The smoothness and strong-convexity assumption is the standard assumption in the literature of analyzing LMC algorithm (Dalalyan, 2017a;b; Cheng & Bartlett, 2018; Durmus et al., 2019; Durmus & Moulines, 2019).

**A 2.** (***Linear Growth of the 3rd-order Derivative***) *Assume $f \in \mathcal{C}^3$ and the operator $\nabla(\Delta f)$ grows at most linearly, i.e., there exists a constant $G > 0$ such that $\|\nabla(\Delta f(\boldsymbol{x}))\| \leq G(1 + \|\boldsymbol{x}\|)$.*

**Remark.** *The linear growth (at infinity) condition on $\nabla\Delta f$ is actually not as restrictive as it appears, and in some sense even weaker than some classical condition for the existence of solutions to SDE. For example, a standard condition for ensuring the existence and uniqueness of a global solution to SDE is at most a linear growth (at infinity) of the drift (Pavliotis, 2014, Theorem 3.1). If we consider monomial potentials, i.e., $f(x) = x^p, p \in \mathbb{N}_+$, then the linear growth condition on $\nabla\Delta f$ is met when $p \leq 4$, whereas the classical condition for the existence of solutions holds only when $p \leq 2$.*

**Remark.** *Another additional assumption, namely Hessian Lipschitz condition, is commonly used in the literature (e.g., Durmus & Moulines (2019); Ma et al. (2021)). It requires the existence of a constant $\tilde{L}$, such that $\|\nabla^2 f(\boldsymbol{y}) - \nabla^2 f(\boldsymbol{x})\| \leq \tilde{L}\|\boldsymbol{y} - \boldsymbol{x}\|$. It can be shown that smoothness and Hessian Lipschitzness imply A2. Meanwhile, examples that satisfy A2 but are not Hessian Lipschitz exist, e.g., $f(x) = x^4$, and thus A2 is not necessarily stronger than Hessian Lipschitzness.*

**Remark.** *Same as $L$ and $m$ in A1, we implicitly assume the constant $G$ introduced in A2 to be independent of dimension. Meanwhile, it is important to note examples for which $G$ depends on the dimension do exist, and this is also true for other regularity constants including not only $L$ and $m$ but also the Hessian Lipschitz constant $\tilde{L}$. This part of the assumption is a strong one.*

**Remark.** *A2, together with Ito's lemma, helps establish an order $p_1 = 2$ of local weak error for LMC (see Lemma D.2), which enables us to obtain the $\sqrt{d}$ dependence.*

To apply mean-square analysis to study LMC algorithm, we will need to ensure the underlying Langevin dynamics is contractive, which we verify in Section C and D in the appendix. In addition, we work out all required constants to determine the $C$ in Eq. 7 explicitly in the appendix. With all these necessary ingredients, we now invoke Theorem 3.4 and obtain the following result:

**Theorem 4.1.** *(**Non-Asymptotic Error Bound: LMC**) Suppose Assumption 1 and 2 hold. LMC iteration $\bar{\boldsymbol{x}}_{k+1} = \bar{\boldsymbol{x}}_k - h\nabla f(\bar{\boldsymbol{x}}_k) + \sqrt{2h}\xi_k$ satisfies*

$$W_2(Law(\bar{\boldsymbol{x}}_k), \mu) \le e^{-mkh} W_2(Law(\boldsymbol{x}_0), \mu) + C_{LMC}h, \quad 0 < h \le \frac{1}{4\kappa L}, k \in \mathbb{N} \qquad (10)$$

*where $C_{LMC} = \frac{10(L^2+G)}{m^{\frac{3}{2}}}\sqrt{2d + m\left(\mathbb{E}\left\|\boldsymbol{x}_0\right\|^2 + 1\right)} = \mathcal{O}(\sqrt{d})$.*

Corollary 3.5 combined with the above result gives the following bound on the mixing time of LMC:

**Theorem 4.2.** *(**Upper Bound of Mixing Time: LMC**) Suppose Assumption 1 and 2 hold. If running LMC from $\boldsymbol{x}_0$, we then have*

$$\tau_{\text{mix}}(\epsilon; W_2; \text{LMC}) \le \max\left\{4\kappa^2, \frac{2C_{LMC}}{m}\frac{1}{\epsilon}\right\} \log \frac{2W_2(Law(\boldsymbol{x}_0), \mu)}{\epsilon}$$

*where $C_{LMC}$ is the same in Theorem 4.1. When high accuracy is needed, i.e., $\epsilon \le \frac{C_{LMC}}{2m\kappa^2}$, we have*

$$\tau_{\text{mix}}(\epsilon; W_2; \text{LMC}) \le \frac{2C_{LMC}}{m}\frac{1}{\epsilon} \log \frac{2W_2(Law(\boldsymbol{x}_0), \mu)}{\epsilon} = \widetilde{\mathcal{O}}\left(\frac{\sqrt{d}}{\epsilon}\right).$$

The $\widetilde{\mathcal{O}}\left(\frac{\sqrt{d}}{\epsilon}\right)$ mixing time bound in $W_2$ distance improves upon the previous ones (Dalalyan, 2017a; Cheng & Bartlett, 2018; Durmus & Moulines, 2019; Durmus et al., 2019) in the dependence of $d$ and/or $\epsilon$. If further assuming $G = \mathcal{O}(L^2)$, we then have $C_{\text{LMC}} = \mathcal{O}(\kappa^2\sqrt{m}\sqrt{d})$ and Thm.4.2 shows the mixing time is $\widetilde{\mathcal{O}}\left(\frac{\kappa^2}{\sqrt{m}}\frac{\sqrt{d}}{\epsilon}\right)$, which also improves the $\kappa$ dependence in some previous results (Dalalyan, 2017a; Cheng & Bartlett, 2018) in the $m \le 1$ regime. A brief summary is in Table 1.

Table 1: Comparison of mixing time results in 2-Wasserten distance of LMC with $L$-smooth and $m$-strongly-convex potential. Constant step size is used and accuracy tolerance $\epsilon$ is small enough.

|  | mixing time | Additional Assumption |
|---|---|---|
| (Dalalyan, 2017a, Theorem 1) | $\widetilde{\mathcal{O}}\left(\frac{\kappa^2}{m} \cdot \frac{d}{\epsilon^2}\right)$ | N/A |
| (Cheng & Bartlett, 2018, Theorem 1) | $\widetilde{\mathcal{O}}\left(\frac{\kappa^2}{m} \cdot \frac{d}{\epsilon^2}\right)$ | N/A |
| (Durmus et al., 2019, Corollary 10) | $\widetilde{\mathcal{O}}\left(\frac{\kappa}{m} \cdot \frac{d}{\epsilon^2}\right)$ | N/A |
| (Durmus & Moulines, 2019, Theorem 8) | $\widetilde{\mathcal{O}}\left(\frac{d}{\epsilon}\right)$ [1] | $\left\|\nabla^2 f(\boldsymbol{x}) - \nabla^2 f(\boldsymbol{y})\right\| \le \widetilde{L}\left\|\boldsymbol{x} - \boldsymbol{y}\right\|$ |
| This work (Theorem 4.2) | $\widetilde{\mathcal{O}}\left(\frac{\kappa^2}{\sqrt{m}} \cdot \frac{\sqrt{d}}{\epsilon}\right)$ | Assumption 2 and $G = \mathcal{O}(L^2)$ [2] |

**Remark** (more comparison). *The seminal work of Li et al. (2019) provided mean-square analysis (their Thm.1) and obtained a $\widetilde{\mathcal{O}}\left(\frac{d}{\epsilon}\right)$ mixing time bound for LMC (their Example 1) under smoothness, strong convexity and Hessian Lipschitz conditions, consistent with that in Durmus & Moulines (2019). By using our version (Thm.3.3) and tracking down constants' dimension-dependence, we are able to tighten it to $\widetilde{\mathcal{O}}\left(\frac{\sqrt{d}}{\epsilon}\right)$. Worth clarifying is, dimension-dependence might not be the focus of Li et al. (2019); instead, it considered $\epsilon$-dependence and other discretizations, and showed for example that 1.5 SRK discretization has improved mixing time bound of $\widetilde{\mathcal{O}}\left(\frac{d}{\epsilon^{2/3}}\right)$. The dimension dependence of this discretization, for example, can possibly be improved by our results too.*

---

[1] The dependence on $\kappa$ is not readily available from Theorem 8 in Durmus & Moulines (2019).

[2] The $G = \mathcal{O}(L^2)$ assumption is only for $\kappa, m$ dependence. Removing it doesn't affect $d, \epsilon$ dependence.

**Optimality** In fact, the $\widetilde{\mathcal{O}}\left(\frac{\sqrt{d}}{\epsilon}\right)$ mixing time of LMC has the optimal scaling one can expect. This is in terms of dependence on $d$ and $\epsilon$, over the class of all log-smooth and log-strongly-convex target measures. To illustrate this, consider the following Gaussian target distribution whose potential is

$$f(\boldsymbol{x}) = \frac{m}{2} \sum_{i=1}^{d} x_i^2 + \frac{L}{2} \sum_{i=d+1}^{2d} x_i^2, \quad \text{with } m = 1, L \geq 4m. \tag{11}$$

We now establish a lower bound on the mixing time of LMC algorithm for this target measure.

**Theorem 4.3.** *(**Lower Bound of Mixing Time**) Suppose we run LMC for the target measure defined in Eq.* (11) *from $\boldsymbol{x}_0 = \boldsymbol{1}_{2d}$, then for any choice of step size $h > 0$ within stability limit, we have*

$$\tau_{\mathrm{mix}}(\epsilon; W_2; \mathrm{LMC}) \geq \frac{\sqrt{d}}{8\epsilon} \log \frac{\sqrt{d}}{\epsilon} = \widetilde{\Omega}\left(\frac{\sqrt{d}}{\epsilon}\right).$$

Combining Theorem 4.2 and 4.3, we see that mean-square analysis provides a tight bound for LMC and $\widetilde{\mathcal{O}}\left(\frac{\sqrt{d}}{\epsilon}\right)$ is the optimal scaling of LMC for target measures satisfying Assumptions 1 and 2.

Note that the above optimality results only partially, but *not* completely, close the gap between the upper and lower bounds of LMC over the entire family of log-smooth and log-strongly-convex target measures, because of one limitation of our result — A(ssumption)2 is, despite of its close relation to the Hessian Lipschitz condition frequently used in the literature, still an extra condition. We tend to believe that A2 may not be essential, but rather an artifact of our proof technique. However, at this moment we cannot eliminate the possibility that the best scaling one can get out of Assumption 1 only (no A2) is worse than $\widetilde{\mathcal{O}}\left(\sqrt{d}/\epsilon\right)$. We'd like to further investigate this in future work.

**Discussion** Besides Li et al. (2019) (see the previous Remark), let's briefly discuss three more important sampling algorithms related to LMC. Two of them are Kinetic Langevin Monte Carlo (KLMC) and Randomized Midpoint Algorithm (RMA), both of which are discretizations of kinetic Langevin dynamics. The other is Metropolis-Adjusted Langevin Algorithm (MALA) which uses the one-step update of LMC as a proposal and then accepts/rejects it with Metropolis-Hastings.

The $\widetilde{\mathcal{O}}\left(\frac{\sqrt{d}}{\epsilon}\right)$ mixing time in 2-Wasserstein distance of KLMC has been established for log-smooth and log-strongly-convex target measures in existing literature (Cheng et al., 2018b; Dalalyan & Riou-Durand, 2020) and that was a milestone. Due to its better dimension dependence over previously best known results of LMC, KLMC is understood to be the analog of Nesterov's accelerated gradient method for sampling (Ma et al., 2021). Our findings show that LMC is able to achieve the same mixing time, although under an additional growth-at-infinity condition. However, this does not say anything about whether/how KLMC accelerates LMC, as the existing KLMC bound may still be not optimal. We also note KLMC has better condition number dependence than our current LMC result, although the $\kappa$ dependence in our bound may not be tight.

RMA (Shen & Lee, 2019) is based on a brilliant randomized discretization of kinetic Langevin dynamics and shown to have further improved dimension dependence (and other pleasant properties). From the perspective of this work, we think it is because RMA is able to break the strong order barrier due to the randomization, and more investigations based on mean-square analysis should be possible.

For MALA, a recent breakthrough (Chewi et al., 2021) establishes a $\widetilde{\mathcal{O}}\left(\sqrt{d}\right)$ mixing time in $W_2$ distance with warm start, and the dimension dependence is shown to be optimal. We see that without the Metropolis adjustment, LMC (under additional assumptions such as A2) can also achieve the same dimension dependence as MALA. But unlike LMC, MALA only has logarithmic dependence on $\frac{1}{\epsilon}$. With warm-start, is it possible/how to improve the dependence of $\frac{1}{\epsilon}$ for LMC, from polynomial to logarithmic? This question is beyond the scope of this paper but worth further investigation.

## 5 NUMERICAL EXAMPLES

This section numerically verifies our theoretical findings for LMC in Section 4, with a particular focus on the dependence of the discretization error in Theorem 4.1 on dimension $d$ and step size $h$.

To this end, we consider two target measures specified by the following two potentials:

$$f_1(\boldsymbol{x}) = \frac{1}{2}\|\boldsymbol{x}\|^2 + \log\left(\sum_{i=1}^{d} e^{x_i}\right) \quad \text{and} \quad f_2(\boldsymbol{x}) = \frac{1}{2}\|\boldsymbol{x}\|^2 - \frac{1}{2d^{\frac{1}{2}}}\sum_{i=1}^{d}\cos\left(d^{\frac{1}{4}}x_i\right). \tag{12}$$

It is not hard to see $f_1$ is 2-smooth and 1-strongly convex, $f_2$ is $\frac{3}{2}$-smooth and 1-strongly-convex, and both satisfy Assumption 2. $f_2$ is also used in (Chewi et al., 2021) to illustrate the optimal dimension dependence of MALA. Explicit expression of 2-Wasserstein distance between non-Gaussian distributions is typically not available, instead, we use the Euclidean norm of the mean error as a surrogate because $\left\|\mathbb{E}\bar{\boldsymbol{x}}_k - \mathbb{E}_\mu\boldsymbol{x}\right\| \leq W_2(\text{Law}(\bar{\boldsymbol{x}}_k), \mu)$ due to Jensen's inequality. To obtain an accurate estimate of the ground truth, we run $10^8$ independent LMC realizations using a tiny step size (h = 0.001), each till a fixed, long enough time, and use the empirical average to approximate $\mathbb{E}_\mu\boldsymbol{x}$.

To study the dimension dependence of sampling error, we fix step size $h = 0.1$, and for each $d \in \{1, 2, 5, 10, 20, 50, 100, 200, 500, 1000\}$, we simulate $10^4$ independent Markov chains using LMC algorithm for 100 iterations, which is long enough for the chain to be well-mixed. The mean and the standard deviation of the sampling error corresponding to the last 10 iterates are recorded.

To study step size dependence of sampling error, we fix $d = 10$ and experiment with step size $h \in \{1, 2, 3, 4, 5, 6, 7, 8, 9, 10\} \times 10^{-1}$. We run LMC till $T = 20$, i.e., $\lceil\frac{T}{h}\rceil$ iterations for each $h$. The procedure is repeated $10^4$ times with different random seeds to obtain independent samples. When the corresponding continuous time $t = kh > 10$, we see from Eq. (10) that LMC is well converged and the sampling error is saturated by the discretization error. Therefore, for each $h$, we take the last $\lceil\frac{10}{h}\rceil$ iterates and record the mean and standard deviation of their sampling error.

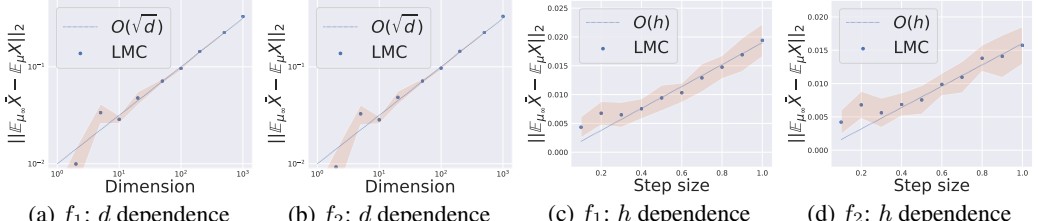

(a) $f_1$: $d$ dependence      (b) $f_2$: $d$ dependence      (c) $f_1$: $h$ dependence      (d) $f_2$: $h$ dependence

Figure 1: Dependence of the sampling error of LMC on dimension $d$ and step size $h$ for $f_1$ and $f_2$. Both axes in Fig.1a & 1b are in log scale. Shaded areas in Fig.1a & 1b represent one std. of the last 10 iterations. Shaded areas in Fig.1c & 1d represent one std. of the last $\lceil\frac{10}{h}\rceil$ iterations.

Results shown in Fig.1 are consistent with our theoretical analysis of the sampling error. Both linear dependence on $\sqrt{d}$ and $h$ can be supported by the empirical evidence. Note results with smaller $h$ are less accurate because one starts to see the error of empirical approximation due to finite samples.

## 6   CONCLUSION

Via a refined mean-square analysis of Langevin Monte Carlo algorithm, we obtain an improved and optimal $\widetilde{\mathcal{O}}\left(\sqrt{d}/\epsilon\right)$ bound on its mixing time, which was previously thought to be obtainable only with the addition of momentum. This was under the standard smoothness and strongly-convexity assumption, plus an addition linear growth condition on the third-order derivative of the potential function, similar to Hessian Lipschitz condition already popularized in the literature.

Here are some possible directions worth further investigations. (i) Combine mean-square analysis with stochastic gradient analysis to study SDE-based stochastic gradient MCMC methods; (ii) Is it still possible to obtain $\sqrt{d}$-dependence without A2, i.e., only under log-smooth and log-strongly-convex conditions? (iii) Applications of mean-square analysis to other SDEs and/or discretizations; (iv) Motivated by Chewi et al. (2021), it would be interesting to know whether the dependence on $\frac{1}{\epsilon}$ can be improved to logarithmic, for example if LMC is initialized at a warm start.

## ACKNOWLEDGMENTS

The authors thank Andre Wibisono, Chenxu Pang, anonymous reviewers and area chair for suggestions that significantly improved the quality of this paper. MT was partially supported by NSF DMS-1847802 and ECCS-1936776. This work was initiated when HZ was a professor and RL was a PhD student at Georgia Institute of Technology.

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

# A   PROOF OF RESULTS IN SECTION 3

## A.1   PROOF OF THEOREM 3.3 (GLOBAL INTEGRATION ERROR, INFINITE TIME VERSION)

*Proof.* We write the solution of an SDE by $\boldsymbol{x}_{t_0,\boldsymbol{x}_{t_0}}(t_0 + t)$ when the dependence on initialization needs highlight. Denote $t_k = kh$ and $\boldsymbol{x}_{t_k} = \boldsymbol{x}_k$ for better readability.

We will first make an easy observation that contraction and bounded 2nd-moment of the invariant distribution lead to bounded 2nd-moment of the SDE solution for all time: let $\boldsymbol{y}_0$ be a random variable following the invariant distribution of Eq. (1), i.e., $\boldsymbol{y}_0 \sim \mu$, then $\boldsymbol{y}_t \sim \mu$ and

$$
\begin{aligned}
\mathbb{E}\left\|\boldsymbol{x}_t\right\|^2 &\leq 2\mathbb{E}\left\|\boldsymbol{x}_t - \boldsymbol{y}_t\right\|^2 + 2\mathbb{E}\left\|\boldsymbol{y}_t\right\|^2 \\
&\leq 2\mathbb{E}\left\|\boldsymbol{x}_0 - \boldsymbol{y}_0\right\|^2 \exp(-2\beta t) + 2\mathbb{E}\left\|\boldsymbol{y}_t\right\|^2 \\
&\leq 4\mathbb{E}(\left\|\boldsymbol{x}_0\right\|^2 + \left\|\boldsymbol{y}_0\right\|^2)\exp(-2\beta t) + 2\mathbb{E}\left\|\boldsymbol{y}_t\right\|^2 \\
&= 4\mathbb{E}\left\|\boldsymbol{x}_0\right\|^2 \exp(-2\beta t) + \left(2 + 4\exp(-2\beta t)\right)\mathbb{E}_{\boldsymbol{y}\sim\mu}\left\|\boldsymbol{y}\right\|^2 \\
&\leq 4\mathbb{E}\left\|\boldsymbol{x}_0\right\|^2 + 6\int_{\mathbb{R}^d}\left\|\boldsymbol{y}\right\|^2 d\mu \triangleq U^2
\end{aligned}
$$

and then it follows that

$$
\mathbb{E}\left\|\bar{\boldsymbol{x}}_k\right\|^2 \leq 2\mathbb{E}\left\|\bar{\boldsymbol{x}}_k - \boldsymbol{x}_k\right\|^2 + 2\mathbb{E}\left\|\boldsymbol{x}_k\right\|^2 \leq 2e_k^2 + 2U^2. \tag{13}
$$

Denote $\langle\boldsymbol{x}, \boldsymbol{y}\rangle_A = \langle A\boldsymbol{x}, A\boldsymbol{y}\rangle$, $\left\|\boldsymbol{x}\right\|_A = \left\|A\boldsymbol{x}\right\|$ and

$$
f_k = \left\{\mathbb{E}\left\|\boldsymbol{x}_k - \bar{\boldsymbol{x}}_k\right\|_A^2\right\}^{\frac{1}{2}} \tag{14}
$$

where $A$ is the non-singular matrix from Equation (4). Also denote that largest and smallest singular values of $A$ by $\sigma_{\max}$ and $\sigma_{\min}$, respectively, and the condition number of $A$ by $\kappa_A = \frac{\sigma_{\max}}{\sigma_{\min}}$. Recall $e_k = \mathbb{E}\left\|\boldsymbol{x}_k - \bar{\boldsymbol{x}}_k\right\|$, it is easy to see that

$$
\sigma_{\min}e_k \leq f_k \leq \sigma_{\max}e_k. \tag{15}
$$

Further, we have the following decomposition

$$
\begin{aligned}
f_{k+1}^2 &= \mathbb{E}\left\|\boldsymbol{x}_{k+1} - \bar{\boldsymbol{x}}_{k+1}\right\|_A^2 \\
&= \mathbb{E}\left\|\boldsymbol{x}_{t_k,\boldsymbol{x}_{t_k}}(t_{k+1}) - \boldsymbol{x}_{t_k,\bar{\boldsymbol{x}}_k}(t_{k+1}) + \boldsymbol{x}_{t_k,\bar{\boldsymbol{x}}_k}(t_{k+1}) - \bar{\boldsymbol{x}}_{k+1}\right\|_A^2 \\
&= \underbrace{\mathbb{E}\left\|\boldsymbol{x}_{t_k,\boldsymbol{x}_{t_k}}(t_{k+1}) - \boldsymbol{x}_{t_k,\bar{\boldsymbol{x}}_k}(t_{k+1})\right\|_A^2}_{\textcircled{1}} + \underbrace{\mathbb{E}\left\|\boldsymbol{x}_{t_k,\bar{\boldsymbol{x}}_k}(t_{k+1}) - \bar{\boldsymbol{x}}_{k+1}\right\|_A^2}_{\textcircled{2}} \\
&\quad + 2\underbrace{\mathbb{E}\langle A\left(\boldsymbol{x}_{t_k,\boldsymbol{x}_{t_k}}(t_{k+1}) - \boldsymbol{x}_{t_k,\bar{\boldsymbol{x}}_k}(t_{k+1})\right), A\left(\boldsymbol{x}_{t_k,\bar{\boldsymbol{x}}_k}(t_{k+1}) - \bar{\boldsymbol{x}}_{k+1}\right)\rangle}_{\textcircled{3}}.
\end{aligned} \tag{16}
$$

Term $\textcircled{1}$ is taken care of the contraction property

$$
\mathbb{E}\left\|\boldsymbol{x}_{t_k,\boldsymbol{x}_{t_k}}(t_{k+1}) - \boldsymbol{x}_{t_k,\bar{\boldsymbol{x}}_k}(t_{k+1})\right\|_A^2 \leq f_k^2 \exp(-2\beta h). \tag{17}
$$

Term $\textcircled{2}$ is dealt with by the bound on local strong error

$$
\mathbb{E}\left\|\boldsymbol{x}_{t_k,\bar{\boldsymbol{x}}_k}(t_{k+1}) - \bar{\boldsymbol{x}}_{k+1}\right\|_A^2 \leq \sigma_{\max}^2\left(C_2^2 + D_2^2\mathbb{E}\left\|\bar{\boldsymbol{x}}_k\right\|^2\right)h^{2p_2}. \tag{18}
$$

Term $\boxed{3}$ requires more efforts to cope with, and by the decomposition in Eq. (5) we have

$$
\begin{aligned}
&\mathbb{E}\langle(\boldsymbol{x}_{t_k,\boldsymbol{x}_{t_k}}(t_{k+1}) - \boldsymbol{x}_{t_k,\bar{\boldsymbol{x}}_k}(t_{k+1}), \boldsymbol{x}_{t_k,\bar{\boldsymbol{x}}_k}(t_{k+1}) - \bar{\boldsymbol{x}}_{k+1}\rangle_A \\
=&\mathbb{E}\langle \boldsymbol{x}_k - \bar{\boldsymbol{x}}_k, \boldsymbol{x}_{t_k,\bar{\boldsymbol{x}}_k}(t_{k+1}) - \bar{\boldsymbol{x}}_{k+1}\rangle_A + \mathbb{E}\langle \boldsymbol{z}_h(\boldsymbol{x}_k,\bar{\boldsymbol{x}}_k), \boldsymbol{x}_{t_k,\bar{\boldsymbol{x}}_k}(t_{k+1}) - \bar{\boldsymbol{x}}_{k+1}\rangle_A \\
\stackrel{(i)}{=}&\mathbb{E}\langle \boldsymbol{x}_k - \bar{\boldsymbol{x}}_k, \mathbb{E}[\boldsymbol{x}_{t_k,\bar{\boldsymbol{x}}_k}(t_{k+1}) - \bar{\boldsymbol{x}}_{k+1}|\mathcal{F}_k]\rangle_A + \mathbb{E}\langle \boldsymbol{z}_h(\boldsymbol{x}_k,\bar{\boldsymbol{x}}_k), \boldsymbol{x}_{t_k,\bar{\boldsymbol{x}}_k}(t_{k+1}) - \bar{\boldsymbol{x}}_{k+1}\rangle_A \\
\stackrel{(ii)}{\leq}&f_k \left(\mathbb{E}\left\|\mathbb{E}[\boldsymbol{x}_{t_k,\bar{\boldsymbol{x}}_k}(t_{k+1}) - \bar{\boldsymbol{x}}_{k+1}|\mathcal{F}_k]\right\|_A^2\right)^{\frac{1}{2}} + \left(\mathbb{E}\left\|\boldsymbol{z}_h(\boldsymbol{x}_k,\bar{\boldsymbol{x}}_k)\right\|_A^2\right)^{\frac{1}{2}} \left(\mathbb{E}\left\|\boldsymbol{x}_{t_k,\bar{\boldsymbol{x}}_k}(t_{k+1}) - \bar{\boldsymbol{x}}_{k+1}\right\|_A^2\right)^{\frac{1}{2}} \\
\stackrel{(iii)}{\leq}&\sigma_{\max}f_k \left(\mathbb{E}\left\|\mathbb{E}[\boldsymbol{x}_{t_k,\bar{\boldsymbol{x}}_k}(t_{k+1}) - \bar{\boldsymbol{x}}_{k+1}|\mathcal{F}_k]\right\|^2\right)^{\frac{1}{2}} + \sigma_{\max}^2 \left(\mathbb{E}\left\|\boldsymbol{z}_h(\boldsymbol{x}_k,\bar{\boldsymbol{x}}_k)\right\|^2\right)^{\frac{1}{2}} \left(\mathbb{E}\left\|\boldsymbol{x}_{t_k,\bar{\boldsymbol{x}}_k}(t_{k+1}) - \bar{\boldsymbol{x}}_{k+1}\right\|^2\right)^{\frac{1}{2}} \\
\stackrel{(iv)}{\leq}&\sigma_{\max}f_k \left(C_1 + D_1\sqrt{\mathbb{E}\left\|\bar{\boldsymbol{x}}_k\right\|^2}\right)h^{p_1} + \kappa_A\sigma_{\max}C_0 f_k\sqrt{h}\left(C_2 + D_2\sqrt{\mathbb{E}\left\|\bar{\boldsymbol{x}}_k\right\|^2}\right)h^{p_2} \\
\stackrel{(v)}{\leq}&\kappa_A\sigma_{\max}(C_1 + C_0C_2)e_kh^{p_2+\frac{1}{2}} + \kappa_A\sigma_{\max}(D_1 + C_0D_2)\sqrt{\mathbb{E}\left\|\bar{\boldsymbol{x}}_k\right\|^2}f_kh^{p_2+\frac{1}{2}}
\end{aligned}
\tag{19}
$$

where $(i)$ uses the tower property of conditional expectation and $\mathcal{F}_k$ is the filtration at $k$-th iteration, $(ii)$ uses Cauchy-Schwarz inequality, $(iii)$ is due to the relationship between $e_k$ and $f_k$, $(iv)$ is due to local weak error, local strong error and Eq. (5), and $(v)$ is due to $p_1 \geq p_2 + \frac{1}{2}$ and $0 < h \leq h_0 \leq 1$.

Now plug Eq. (17), (18) and (19) in Eq. (16), we obtain

$$
\begin{aligned}
f_{k+1}^2 \leq&f_k^2\exp(-2\beta h) + \sigma_{\max}^2\left(C_2^2 + D_2^2\mathbb{E}\left\|\bar{\boldsymbol{x}}_k\right\|^2\right)h^{2p_2} + \kappa_A\sigma_{\max}(C_1 + C_0C_2)f_kh^{p_2+\frac{1}{2}} \\
&+ \kappa_A\sigma_{\max}(D_1 + C_0D_2)\sqrt{\mathbb{E}\left\|\bar{\boldsymbol{x}}_k\right\|^2}f_kh^{p_2+\frac{1}{2}} \\
\stackrel{(i)}{\leq}&\left(1 - \frac{7}{8}\beta h\right)f_k^2 + \sigma_{\max}^2\left(C_2^2 + D_2^2\mathbb{E}\left\|\bar{\boldsymbol{x}}_k\right\|^2\right)h^{2p_2} + \kappa_A\sigma_{\max}(C_1 + C_0C_2)f_kh^{p_2+\frac{1}{2}} \\
&+ \kappa_A\sigma_{\max}(D_1 + C_0D_2)\sqrt{\mathbb{E}\left\|\bar{\boldsymbol{x}}_k\right\|^2}f_kh^{p_2+\frac{1}{2}} \\
\stackrel{(ii)}{\leq}&\left(1 - \frac{7}{8}\beta h\right)f_k^2 + \kappa_A\sigma_{\max}\left(C_1 + C_0C_2 + \sqrt{2}U(D_1 + C_0D_2)\right)f_kh^{p_2+\frac{1}{2}} + 2\kappa_A^2D_2^2f_k^2h^{2p_2} \\
&+ \sqrt{2}\kappa_A^2(D_1 + C_0D_2)f_k^2h^{p_2+\frac{1}{2}} + \sigma_{\max}^2\left(C_2^2 + 2D_2^2U^2\right)h^{2p_2} \\
\stackrel{(iii)}{\leq}&\left(1 - \frac{7}{8}\beta h\right)f_k^2 + \kappa_A\sigma_{\max}\left(C_1 + C_0C_2 + \sqrt{2}U(D_1 + C_0D_2)\right)f_kh^{p_2+\frac{1}{2}} + \frac{3\beta}{8}f_k^2h \\
&+ \sigma_{\max}^2\left(C_2^2 + 2D_2^2U^2\right)h^{2p_2} \\
=&\left(1 - \frac{1}{2}\beta h\right)f_k^2 + \kappa_A\sigma_{\max}\left(C_1 + C_0C_2 + \sqrt{2}U(D_1 + C_0D_2)\right)f_kh^{p_2+\frac{1}{2}} \\
&+ \sigma_{\max}^2\left(C_2^2 + 2D_2^2U^2\right)h^{2p_2} \\
\stackrel{(iv)}{\leq}&\left(1 - \frac{1}{2}\beta h\right)f_k^2 + \frac{\beta}{4}f_k^2h + \frac{\kappa_A^2\sigma_{\max}^2\left(C_1 + C_0C_2 + \sqrt{2}U(D_1 + C_0D_2)\right)^2}{\beta}h^{2p_2} \\
&+ \sigma_{\max}^2\left(C_2^2 + 2D_2^2U^2\right)h^{2p_2} \\
=&\left(1 - \frac{1}{4}\beta h\right)f_k^2 + \kappa_A^2\sigma_{\max}^2\left(\frac{\left(C_1 + C_0C_2 + \sqrt{2}U(D_1 + C_0D_2)\right)^2}{\beta} + C_2^2 + 2D_2^2U^2\right)h^{2p_2}
\end{aligned}
$$

where $(i)$ is due to the assumption $0 < h \leq \frac{1}{4\beta}$ and $e^{-x} \leq 1 - x + \frac{x^2}{2}$ for $0 < x < 1$, $(ii)$ is due to the upper bound on $\mathbb{E}\left\|\bar{\boldsymbol{x}}_k\right\|^2$ in Eq. (13), $(iii)$ holds provided when $h \leq$

$$\min\left\{\left(\frac{\sqrt{\beta}}{4\sqrt{2}\kappa_A D_2}\right)^{\frac{1}{p_2-\frac{1}{2}}},\left(\frac{\beta}{8\sqrt{2}\kappa_A^2(D_1+C_0 D_2)}\right)^{\frac{1}{p_2-\frac{1}{2}}}\right\}$$ and $(iv)$ is due to Cauchy-Schwarz inequality.

Unfolding the above inequality gives us

$$f_{k+1}^2 \leq \frac{4}{\beta}\kappa_A^2\sigma_{\max}^2\left(\frac{\left(C_1+C_0 C_2+\sqrt{2}U(D_1+C_0 D_2)\right)^2}{\beta}+C_2^2+2D_2^2 U^2\right)h^{2p_2-1}.$$

Taking square root on both sides and using $\sqrt{a^2+b^2+c^2}\leq a+b+c,\forall a,b,c\geq 0$ yields

$$f_{k+1}\leq\frac{2}{\sqrt{\beta}}\kappa_A\sigma_{\max}\left(\frac{C_1+C_0 C_2+\sqrt{2}U(D_1+C_0 D_2)}{\sqrt{\beta}}+C_2+\sqrt{2}D_2 U\right)h^{p_2-\frac{1}{2}}.$$

Finally using the relationship between $e_k$ and $f_k$, we obtain

$$e_k\leq\frac{2}{\sqrt{\beta}}\kappa_A^2\left(\frac{C_1+C_0 C_2+\sqrt{2}U(D_1+C_0 D_2)}{\sqrt{\beta}}+C_2+\sqrt{2}D_2 U\right)h^{p_2-\frac{1}{2}}.$$

$\square$

## A.2 Proof of Theorem 3.4 (Non-Asymptotic Sampling Error Bound: General Case)

*Proof.* Let $\boldsymbol{y}_0\sim\mu$ and $(\boldsymbol{x}_0,\boldsymbol{y}_0)$ are coupled such that $\mathbb{E}\|\boldsymbol{x}_0-\boldsymbol{y}_0\|^2=W_2^2(\mathrm{Law}(\boldsymbol{x}_0),\mu)$. Denote the solution of Eq. (1) starting from $\boldsymbol{x}_0,\boldsymbol{y}_0$ by $\boldsymbol{x}_t,\boldsymbol{y}_t$ respectively, and $t_k=kh$. We have

$$\begin{aligned}
W_2(\mathrm{Law}(\bar{\boldsymbol{x}}_k),\mu)&\leq W_2(\mathrm{Law}(\bar{\boldsymbol{x}}_k),\mathrm{Law}(\boldsymbol{x}_{t_k}))+W_2(\mathrm{Law}(\boldsymbol{x}_{t_k}),\mu)\\
&\leq\sqrt{\mathbb{E}\left\|\bar{\boldsymbol{x}}_k-\boldsymbol{x}_{t_k}\right\|^2}+\sqrt{\mathbb{E}\left\|\boldsymbol{x}_{t_k}-\boldsymbol{y}_{t_k}\right\|^2}\\
&\overset{(i)}{\leq}e_k+\sqrt{\mathbb{E}\|\boldsymbol{x}_0-\boldsymbol{y}_0\|^2\exp\left(-2\beta t_k\right)}\\
&=e_k+\exp\left(-\beta t_k\right)W_2(\mathrm{Law}(\boldsymbol{x}_0),\mu)
\end{aligned}$$

where $(i)$ is due to the contraction assumption on Eq. (1). Invoking the conclusion of Theorem 3.3 completes the proof. $\square$

## A.3 Proof of Corollary 3.5 (Upper Bound of Mixing Time: General Case)

*Proof.* Given any tolerance $\epsilon>0$, we know from Theorem 3.4 that if $k$ is large enough and $h$ is small enough such that

$$\exp\left(-\beta kh\right)W_2(\mathrm{Law}(\boldsymbol{x}_0),\mu)\leq\frac{\epsilon}{2}. \tag{20}$$

$$Ch^{p_2-\frac{1}{2}}\leq\frac{\epsilon}{2} \tag{21}$$

we then have $W_2(\mathrm{Law}(\bar{\boldsymbol{x}}_k),\mu)\leq\epsilon$. Solving Inequality (20) yields

$$k\geq\frac{1}{\beta h}\log\frac{2W_2(\mathrm{Law}(\boldsymbol{x}_0),\mu)}{\epsilon}\triangleq k^{\star} \tag{22}$$

To minimize the lower bound, we want pick step size $h$ as large as possible. Besides $h\leq h_1$, Eq. (21) poses further constraint on $h$, hence we have

$$h\leq\min\left\{h_1,\left(\frac{\epsilon}{2C}\right)^{\frac{1}{p_2-\frac{1}{2}}}\right\}.$$

Plug the upper bound of $h$ in Eq. (22), we have

$$k^\star = \max\left\{\frac{1}{\beta h_1}, \frac{1}{\beta}\left(\frac{2C}{\epsilon}\right)^{\frac{1}{p_2-\frac{1}{2}}}\right\} \log\frac{2W_2(\text{Law}(\boldsymbol{x}_0), \mu)}{\epsilon}.$$

When high accuracy is needed, i.e., $\epsilon < 2Ch_1^{p_2-\frac{1}{2}}$, we have

$$k^\star = \frac{(2C)^{\frac{1}{p_2-\frac{1}{2}}}}{\beta}\frac{1}{\epsilon^{\frac{1}{p_2-\frac{1}{2}}}}\log\frac{2W_2(\text{Law}(\boldsymbol{x}_0), \mu)}{\epsilon} = \widetilde{\mathcal{O}}\left(\frac{C^{\frac{1}{p_2-\frac{1}{2}}}}{\beta}\frac{1}{\epsilon^{\frac{1}{p_2-\frac{1}{2}}}}\right).$$

$\square$

# B  PROOF OF RESULTS IN SECTION 4

## B.1  PROOF OF THEOREM 4.1 (NON-ASYMPTOTIC ERROR BOUND: LMC)

*Proof.* From Lemma C.1 we know that Langevin dynamics is a member of the family of contractive SDE, and with a contraction rate of strong-convexity coefficient $\beta = m$ (w.r.t. identity matrix $I_{d\times d}$).

Next, we will need to work out the constants $C_0, C_1, D_1, D_2, C_2$ needed in Theorem 3.3. We have $C_0 = \frac{\sqrt{m}}{2}$, implied from Lemma C.3.

The local strong error and local weak error are bounded in Lemma D.1 and D.2 respectively. Note that the coefficient $\widetilde{C}_1/\widetilde{C}_2$ in the bound for local strong/weak error depends on initial value, which changes from iteration to iteration. Combined with Lemma D.3, we would obtain $C_1$ and $C_2$, namely

$$\widetilde{C}_1 \le 2(L^2 + G)\left(\frac{d}{4\kappa L} + \mathbb{E}\|\boldsymbol{x}_0\|^2 + \frac{8d}{7m} + 1\right)^{\frac{1}{2}} \le 2(L^2 + G)\sqrt{\frac{2d}{m} + \mathbb{E}\|\boldsymbol{x}_0\|^2 + 1} \triangleq C_1$$

and

$$\widetilde{C}_2 \le 2L\left(d + \frac{m}{2}\left(\mathbb{E}\|\boldsymbol{x}_0\|^2 + \frac{8d}{7m}\right)\right)^{\frac{1}{2}} \le 2L\sqrt{m}\sqrt{\frac{2d}{m} + \mathbb{E}\|\boldsymbol{x}_0\|^2 + 1} \triangleq C_2.$$

We collect all constants here in the proof for easier reference

$$A = I_{d\times d}, \kappa_A = 1, \beta = m, h_0 = \frac{1}{4\kappa L}, C_0 = \frac{\sqrt{m}}{2},$$

$$C_1 = 2(L^2 + G)\sqrt{\frac{2d}{m} + \mathbb{E}\|\boldsymbol{x}_0\|^2 + 1}, D_1 = 0$$

$$C_2 = 2L\sqrt{m}\sqrt{\frac{2d}{m} + \mathbb{E}\|\boldsymbol{x}_0\|^2 + 1}, D_2 = 0.$$

Then the constant in Theorem 3.3 for LMC algorithm simplifies to

$$C = \frac{2}{\sqrt{\beta}}\left(\frac{C_1 + C_0 C_2}{\sqrt{\beta}} + C_2\right),$$

$$\le \frac{10(L^2 + G)}{m^{\frac{3}{2}}}\sqrt{2d + m\left(\mathbb{E}\|\boldsymbol{x}_0\|^2 + 1\right)} \triangleq C_{\text{LMC}}.$$

Assuming $L, m, G$ are all constants and independent of $d$, then clearly $C_{\text{LMC}} = \mathcal{O}(\sqrt{d})$. Then applying Theorem 3.4 to LMC, we have

$$W_2(\text{Law}(\bar{\boldsymbol{x}}_k), \mu) \le e^{-mkh}W_2(\text{Law}(\boldsymbol{x}_0), \mu) + C_{\text{LMC}}h \tag{23}$$

for $0 < h \le \frac{1}{4\kappa L}$.

$\square$

### B.2   PROOF OF THEOREM 4.3 (LOWER BOUND OF MIXING TIME)

*Proof.* If we start from $\boldsymbol{x}_0 = \mathbf{1}_{2d}$ and run LMC for the potential function in Eq. (11), we then have

$$(\bar{\boldsymbol{x}}_k)_i = \begin{cases} (1-mh)^k(\boldsymbol{x}_0)_i + \sqrt{2h}\sum_{l=1}^k (1-mh)^{k-l}(\boldsymbol{\xi}_l)_i,\ 1 \le i \le d \\ (1-Lh)^k(\boldsymbol{x}_0)_i + \sqrt{2h}\sum_{l=1}^k (1-Lh)^{k-l}(\boldsymbol{\xi}_l)_i,\ d+1 \le i \le 2d \end{cases}$$

and hence

$$(\bar{\boldsymbol{x}}_k)_i \sim \begin{cases} \mathcal{N}\left((1-mh)^k, \frac{2}{m(2-mh)}\left(1-(1-mh)^{2k}\right)\right),\ 1 \le i \le d \\ \mathcal{N}\left((1-Lh)^k, \frac{2}{L(2-Lh)}\left(1-(1-Lh)^{2k}\right)\right),\ d+1 \le i \le 2d \end{cases}$$

Clearly, stability requires $h < \frac{2}{L}$.

The squared 2-Wasserstein distance between the law of the $k$-th iterate of LMC and target distribution is

$$W_2^2(\text{Law}(\bar{\boldsymbol{x}}_k), \mu) = d(1-mh)^{2k} + \frac{d}{m}\left(\sqrt{\frac{2}{2-mh}}\sqrt{1-(1-mh)^{2k}} - 1\right)^2$$

$$+ d(1-Lh)^{2k} + \frac{d}{L}\left(\sqrt{\frac{2}{2-Lh}}\sqrt{1-(1-Lh)^{2k}} - 1\right)^2.$$

Suppose $W_2(\text{Law}(\bar{\boldsymbol{x}}_k), \mu) \le \epsilon$, we then must have

$$d(1-mh)^{2k} \le \epsilon^2 \tag{24}$$

$$\frac{d}{m}\left(\sqrt{\frac{2}{2-mh}}\sqrt{1-(1-mh)^{2k}} - 1\right)^2 \le \epsilon^2. \tag{25}$$

A necessary condition of Eq. (25) is that

$$1 + \frac{\sqrt{m}}{\sqrt{d}}\epsilon \ge \sqrt{\frac{2}{2-mh}}\sqrt{1-(1-mh)^{2k}} \overset{(i)}{\ge} \sqrt{\frac{2}{2-mh}}\sqrt{1-\frac{\epsilon^2}{d}} \tag{26}$$

where $(i)$ is due to Eq. (24). It follows from Eq. (26) and $m=1$ that

$$h \le \frac{4}{1+\frac{\epsilon}{\sqrt{d}}}\frac{\epsilon}{\sqrt{d}} \le \frac{4\epsilon}{\sqrt{d}}. \tag{27}$$

Revisiting Eq. (24) yields

$$\epsilon^2 \ge d(1-mh)^{2k} \overset{(i)}{\ge} d\left(1 - 2mh + \frac{(2mh)^2}{2}\right)^{2k} \overset{(ii)}{\ge} de^{-4mkh}$$

$$\iff k \ge \frac{1}{2hm}\log\frac{\sqrt{d}}{\epsilon} \tag{28}$$

where $(i)$ is due to $mh < \frac{2}{\kappa} < \frac{1}{2}$ and $(ii)$ is due to $e^{-x} \le 1 - x + \frac{x^2}{2}, 0 < x < 1$.

Combine Eq. (27) and (28), we then obtain a lower bound of the mixing time

$$k \ge \frac{\sqrt{d}}{8m\epsilon}\log\frac{\sqrt{d}}{\epsilon} = \frac{\sqrt{d}}{8\epsilon}\log\frac{\sqrt{d}}{\epsilon} = \tilde{\Omega}\left(\frac{\sqrt{d}}{\epsilon}\right).$$

$\square$

## C   SOME PROPERTIES OF LANGEVIN DYNAMICS

### C.1   CONTRACTION OF LANGEVIN DYNAMICS

**Lemma C.1.** *Suppose Assumption 1 holds. Then two copies of overdamped Langevin dynamics have the following contraction property*

$$\left\{ \mathbb{E} \left\| \boldsymbol{y}_t - \boldsymbol{x}_t \right\|^2 \right\}^{\frac{1}{2}} \leq \left\{ \mathbb{E} \left\| \boldsymbol{y} - \boldsymbol{x} \right\|^2 \right\}^{\frac{1}{2}} \exp(-mt)$$

*where $\boldsymbol{x}, \boldsymbol{y}$ are the initial values of $\boldsymbol{x}_t, \boldsymbol{y}_t$.*

*Proof.* First assume $\boldsymbol{x}, \boldsymbol{y}$ are deterministic. Suppose $\boldsymbol{x}_t, \boldsymbol{y}_t$ are respectively the solutions to

$$d\boldsymbol{x}_t = -\nabla f(\boldsymbol{x}_t)dt + \sqrt{2}d\boldsymbol{B}_t$$
$$d\boldsymbol{y}_t = -\nabla f(\boldsymbol{y}_t)dt + \sqrt{2}d\boldsymbol{B}_t$$

where $\boldsymbol{B}_t$ is a standard $d$-dimensional Brownian motion. Denote $L_t = \frac{1}{2}\mathbb{E} \left\| \boldsymbol{y}_t - \boldsymbol{x}_t \right\|^2$ and take time derivative, we obtain

$$\frac{d}{dt}L_t = -\mathbb{E}\langle \boldsymbol{y}_t - \boldsymbol{x}_t, \nabla f(\boldsymbol{y}_t) - \nabla f(\boldsymbol{x}_t)\rangle \overset{(i)}{\leq} -m\mathbb{E} \left\| \boldsymbol{y}_t - \boldsymbol{x}_t \right\|^2 = -2mL_t$$

where $(i)$ is due to the strong-convexity assumption made on $f$. We then obtain $L_t \leq L_0 \exp(-2mt)$ and it follows by Gronwall's inequality that

$$\left\{ \mathbb{E} \left\| \boldsymbol{y}_t - \boldsymbol{x}_t \right\|^2 \right\}^{\frac{1}{2}} \leq \left\| \boldsymbol{y} - \boldsymbol{x} \right\| \exp(-mt).$$

When $\boldsymbol{x}, \boldsymbol{y}$ are random, by the conditioning version of the above inequality and Jensen's inequality, we have

$$\left\{ \mathbb{E}\left[ \mathbb{E} \left\| \boldsymbol{y}_t - \boldsymbol{x}_t \right\|^2 \Big| \boldsymbol{x}, \boldsymbol{y} \right] \right\}^{\frac{1}{2}} \leq \left\{ \mathbb{E} \left\| \boldsymbol{y} - \boldsymbol{x} \right\|^2 \exp(-2mt) \right\}^{\frac{1}{2}} = \left\{ \mathbb{E} \left\| \boldsymbol{y} - \boldsymbol{x} \right\|^2 \right\}^{\frac{1}{2}} \exp(-mt).$$

$\square$

### C.2   GROWTH BOUND OF LANGEVIN DYNAMICS

**Lemma C.2.** *Suppose Assumption 1 holds, then when $0 \leq h \leq \frac{1}{4\kappa L}$, the solution of overdamped Langevin dynamics $\boldsymbol{x}_t$ satisfies*

$$\mathbb{E} \left\| \boldsymbol{x}_h - \boldsymbol{x} \right\|^2 \leq 6 \left( d + \frac{m}{2}\mathbb{E} \left\| \boldsymbol{x} \right\|^2 \right) h$$

*where $\boldsymbol{x}$ is the initial value at $t = 0$.*

*Proof.* We have

$$
\begin{aligned}
\mathbb{E}\left\|\boldsymbol{x}_h - \boldsymbol{x}\right\|^2 =& \mathbb{E}\left\|-\int_0^h \nabla f(\boldsymbol{x}_t)dt + \sqrt{2}\int_0^h d\boldsymbol{B}_t\right\|^2 \\
\leq& 2\mathbb{E}\left\|\int_0^h \nabla f(\boldsymbol{x}_t)dt\right\|^2 + 4\mathbb{E}\left\|\int_0^h d\boldsymbol{B}_t\right\|^2 \\
\overset{(i)}{=}& 2\mathbb{E}\left\|\int_0^h \nabla f(\boldsymbol{x}_t)dt\right\|^2 + 4hd \\
\leq& 2\mathbb{E}\left[\left(\int_0^h \left\|\nabla f(\boldsymbol{x}_t) - \nabla f(\boldsymbol{x})\right\|dt + \int_0^h \left\|\nabla f(\boldsymbol{x})\right\|dt\right)^2\right] + 4hd \\
\leq& 2\mathbb{E}\left[\left(L\int_0^h \left\|\boldsymbol{x}_t - \boldsymbol{x}\right\|dt + h\left\|\nabla f(\boldsymbol{x})\right\|\right)^2\right] + 4hd \\
\leq& 4\mathbb{E}\left[L^2\left(\int_0^h \left\|\boldsymbol{x}_t - \boldsymbol{x}\right\|dt\right)^2 + h^2\left\|\nabla f(\boldsymbol{x})\right\|^2\right] + 4hd \\
\overset{(ii)}{\leq}& 4hd + 4h^2\mathbb{E}\left\|\nabla f(\boldsymbol{x})\right\|^2 + 4L^2h\int_0^h \mathbb{E}\left\|\boldsymbol{x}_t - \boldsymbol{x}\right\|^2 dt
\end{aligned}
$$

where $(i)$ is due to Ito's isometry, $(ii)$ is due to Cauchy-Schwarz inequality. By Gronwall's inequality, we obtain

$$
\mathbb{E}\left\|\boldsymbol{x}_h - \boldsymbol{x}\right\|^2 \leq 4h\left(d + h\mathbb{E}\left\|\nabla f(\boldsymbol{x})\right\|^2\right)\exp\left\{4L^2h^2\right\}.
$$

Since $\left\|\nabla f(\boldsymbol{x})\right\| = \left\|\nabla f(\boldsymbol{x}) - \nabla f(\boldsymbol{0})\right\| \leq L\left\|\boldsymbol{x}\right\|$, when $0 < h \leq \frac{1}{4\kappa L}$, we finally reach at

$$
\mathbb{E}\left\|\boldsymbol{x}_h - \boldsymbol{x}\right\|^2 \leq 4e^{\frac{1}{4}}\left(d + 2hL^2\mathbb{E}\left\|\boldsymbol{x}\right\|^2\right)h \leq 6\left(d + \frac{m}{2}\mathbb{E}\left\|\boldsymbol{x}\right\|^2\right)h.
$$

$\square$

### C.3 BOUND ON EVOLVED DEVIATION

**Lemma C.3.** *Suppose Assumption 1 holds. Let $\boldsymbol{x}_t, \boldsymbol{y}_t$ be two solutions of overdamped Langevin dynamics starting from $\boldsymbol{x}, \boldsymbol{y}$ respectively, for $0 < h \leq \frac{1}{4\kappa L}$, we have the following representation*

$$
\boldsymbol{x}_h - \boldsymbol{y}_h = \boldsymbol{x} - \boldsymbol{y} + \boldsymbol{z}
$$

*with*

$$
E\left\|\boldsymbol{z}\right\|^2 \leq \frac{m}{4}\mathbb{E}\left\|\boldsymbol{x} - \boldsymbol{y}\right\|^2 h.
$$

*Proof.* Let $\boldsymbol{z} = (\boldsymbol{x}_h - \boldsymbol{y}_h) - (\boldsymbol{x} - \boldsymbol{y}) = -\int_0^h \nabla f(\boldsymbol{x}_s) - \nabla f(\boldsymbol{y}_s)ds$. Ito's lemma readily implies that

$$
\begin{aligned}
\mathbb{E}\left\|\boldsymbol{x}_h - \boldsymbol{y}_h\right\|^2 =& \mathbb{E}\left\|\boldsymbol{x} - \boldsymbol{y}\right\|^2 - 2\mathbb{E}\int_0^h \langle \boldsymbol{x}_s - \boldsymbol{y}_s, \nabla f(\boldsymbol{x}_s) - \nabla f(\boldsymbol{y}_s)\rangle ds \\
\overset{(i)}{\leq}& \mathbb{E}\left\|\boldsymbol{x} - \boldsymbol{y}\right\|^2 - 2m\int_0^h \mathbb{E}\left\|\boldsymbol{x}_s - \boldsymbol{y}_s\right\|^2 ds \\
\leq& \mathbb{E}\left\|\boldsymbol{x} - \boldsymbol{y}\right\|^2
\end{aligned}
$$

where $(i)$ is due to strong-convexity of $f$. We then have that

$$
\begin{aligned}
\mathbb{E}\left\|\boldsymbol{z}\right\|^2 &= \left\|\mathbb{E}\left[\int_0^h \nabla f(\boldsymbol{x}_s) - \nabla f(\boldsymbol{y}_s) ds\right]\right\|^2 \\
&\leq \left(\int_0^h \left\|\mathbb{E}\left[\nabla f(\boldsymbol{x}_s) - \nabla f(\boldsymbol{y}_s)\right]\right\| ds\right)^2 \\
&\leq \int_0^h 1^2 ds \int_0^h \left\|\mathbb{E}\left[\nabla f(\boldsymbol{x}_s) - \nabla f(\boldsymbol{y}_s)\right]\right\|^2 ds \\
&\leq h\int_0^h \mathbb{E}\left\|\nabla f(\boldsymbol{x}_s) - \nabla f(\boldsymbol{y}_s)\right\|^2 ds \\
&\leq L^2 h \int_0^h \mathbb{E}\left\|\boldsymbol{x}_s - \boldsymbol{y}_s\right\|^2 ds \\
&\leq L^2 \mathbb{E}\left\|\boldsymbol{x} - \boldsymbol{y}\right\|^2 h^2 \\
&\overset{(i)}{\leq} \frac{m}{4}\mathbb{E}\left\|\boldsymbol{x} - \boldsymbol{y}\right\|^2 h
\end{aligned}
$$

where $(i)$ is due to $h \leq \frac{1}{4\kappa L}$. $\qquad\square$

## D  SOME PROPERTIES OF LMC ALGORITHM

### D.1  LOCAL STRONG ERROR

**Lemma D.1.** *Suppose Assumption 1 holds. Denote the one-step iteration of LMC algorithm with step size $h$ by $\bar{\boldsymbol{x}}_1$ and the solution of overdamped Langevin dynamics at time $t = h$ by $\boldsymbol{x}_h$. Both the discrete algorithm and the continuous dynamics start from the same initial value $\boldsymbol{x}$. If $0 \leq h \leq \frac{1}{4\kappa L}$, then the local strong error of LMC algorithm satisfies*

$$
\left\{\mathbb{E}\left\|\bar{\boldsymbol{x}}_1 - \boldsymbol{x}_h\right\|^2\right\}^{\frac{1}{2}} \leq \widetilde{C}_2 h^{\frac{3}{2}}
$$

*with $\widetilde{C}_2 = 2L\left(d + \frac{m}{2}\mathbb{E}\left\|\boldsymbol{x}\right\|^2\right)^{\frac{1}{2}}$.*

*Proof.* We have for $0 \leq h \leq \frac{1}{4\kappa L}$,

$$
\begin{aligned}
\mathbb{E}\left\|\bar{\boldsymbol{x}}_1 - \boldsymbol{x}_h\right\|^2 &= \mathbb{E}\left\|\int_0^h \nabla f(\boldsymbol{x}_s) - \nabla f(\boldsymbol{x}) ds\right\|^2 \\
&\leq \mathbb{E}\left(\int_0^h \left\|\nabla f(\boldsymbol{x}_s) - \nabla f(\boldsymbol{x})\right\| ds\right)^2 \\
&\leq L^2 \mathbb{E}\left(\int_0^h \left\|\boldsymbol{x}_s - \boldsymbol{x}\right\| ds\right)^2 \\
&\overset{(i)}{\leq} L^2 h \int_0^h \mathbb{E}\left\|\boldsymbol{x}_s - \boldsymbol{x}\right\|^2 ds \\
&\overset{(ii)}{\leq} 3L^2\left(d + \frac{m}{2}\mathbb{E}\left\|\boldsymbol{x}\right\|^2\right) h^3
\end{aligned}
$$

where $(i)$ is due to Cauchy-Schwartz inequality and $(ii)$ is due to Lemma C.2. Taking square roots on both side completes the proof. $\qquad\square$

## D.2 Local Weak Error

**Lemma D.2.** *Suppose Assumption 1 and 2 hold. Denote the one-step iteration of LMC algorithm with step size $h$ by $\bar{x}_1$ and the solution of overdamped Langevin dynamics at time $t = h$ by $x_h$. Both the discrete algorithm and the continuous dynamics start from the same initial value $x$. If $0 \leq h \leq \frac{1}{4\kappa L}$, then the local weak error of LMC algorithm satisfies*

$$\|\mathbb{E}\bar{x}_1 - \mathbb{E}x_h\| \leq \widetilde{C}_1 h^2$$

*with $\widetilde{C}_1 = 2(L^2 + G)\left(\frac{d}{4\kappa L} + \mathbb{E}\|x\|^2 + 1\right)^{\frac{1}{2}}$.*

*Proof.* By Ito's lemma, we have

$$d\nabla f(x_t) = -\nabla^2 f(x_t)\nabla f(x_t)dt + \nabla(\Delta f(x_t))dt + \sqrt{2}\int_0^t \nabla^2 f(x_t)dB_t.$$

It follows that

$$\|\mathbb{E}\bar{x}_1 - \mathbb{E}x_h\| = \left\|\mathbb{E}\int_0^h \nabla f(x_s) - \nabla f(x)ds\right\|$$

$$= \left\|\mathbb{E}\left\{\int_0^h \int_0^s -\nabla^2 f(x_r)\nabla f(x_r) + \nabla(\Delta f(x_r))drds + \sqrt{2}\int_0^h \int_0^s \nabla^2 f(x_r)dB_rds\right\}\right\|$$

$$= \left\|\mathbb{E}\left\{\int_0^h \int_0^s -\nabla^2 f(x_r)\nabla f(x_r) + \nabla(\Delta f(x_r))drds\right\}\right\|$$

$$\leq \int_0^h \int_0^s \mathbb{E}\left\|\nabla^2 f(x_r)\nabla f(x_r)\right\|drds + \int_0^h \int_0^s \mathbb{E}\left\|\nabla(\Delta f(x_r))\right\|drds$$

$$\leq L\int_0^h \int_0^s \mathbb{E}\left\|\nabla f(x_r)\right\|drds + \int_0^h \int_0^s \mathbb{E}\left\|\nabla(\Delta f(x_r))\right\|drds$$

$$\overset{(i)}{\leq} (L^2 + G)\int_0^h \int_0^s \mathbb{E}\|x_r\|drds + \frac{G}{2}h^2$$

$$\leq (L^2 + G)\left(\int_0^h \int_0^s \mathbb{E}\|x_r - x\|drds + \frac{h^2}{2}\mathbb{E}\|x\|\right) + \frac{G}{2}h^2$$

$$\overset{(ii)}{\leq} (L^2 + G)\left(\int_0^h \int_0^s \sqrt{\mathbb{E}\|x_r - x\|^2}drds + \frac{h^2}{2}\mathbb{E}\|x\|\right) + \frac{G}{2}h^2$$

$$\overset{(iii)}{\leq} (L^2 + G)\left(\int_0^h \int_0^s \sqrt{6\left(d + \frac{m}{2}\mathbb{E}\|x\|^2\right)}rdrds + \frac{h^2}{2}\mathbb{E}\|x\|\right) + \frac{G}{2}h^2$$

$$= (L^2 + G)\left(\frac{4\sqrt{6}}{15}\sqrt{\left(d + \frac{m}{2}\mathbb{E}\|x\|^2\right)h} + \frac{1}{2}\mathbb{E}\|x\|\right)h^2 + \frac{G}{2}h^2$$

$$\overset{(iv)}{\leq} (L^2 + G)h^2\sqrt{\left(d + \frac{m}{2}\mathbb{E}\|x\|^2\right)h + \frac{1}{2}\mathbb{E}\|x\|^2} + \frac{G}{2}h^2$$

$$\overset{(v)}{\leq} (L^2 + G)h^2\sqrt{\frac{d}{4\kappa L} + \mathbb{E}\|x\|^2} + \frac{G}{2}h^2$$

$$\leq (L^2 + G)\left(\sqrt{\frac{d}{4\kappa L} + \mathbb{E}\|x\|^2} + 1\right)h^2$$

$$\leq 2(L^2 + G)\left(\frac{d}{4\kappa L} + \mathbb{E}\|x\|^2 + 1\right)^{\frac{1}{2}}h^2$$

where $(i)$ is due to Assumption 2, $(ii)$ is due to Jensen's inequality, $(iii)$ is due to Lemma C.2, $(iv)$ is due to $\sqrt{a} + \sqrt{b} \leq \sqrt{2}\sqrt{a^2 + b^2}$ and $(v)$ is due to $h \leq \frac{1}{4\kappa L}$. It is worth noting in the third equation that the Ito's correction term $\nabla \Delta f$ can also be written as $\Delta \nabla f$ as the two operators commute for $\mathcal{C}^3$ functions. $\qquad\square$

### D.3 Boundedness of LMC Algorithm

**Lemma D.3.** *Suppose Assumption 1 holds. Denote the iterates of LMC by $\bar{x}_k$. If $0 \leq h \leq \frac{1}{4\kappa L}$ we then have the iterates of LMC algorithm are uniformly upper bounded by*

$$\mathbb{E}\left\|\bar{x}_k\right\|^2 \leq \mathbb{E}\left\|x_0\right\|^2 + \frac{8d}{7m}, \quad \forall k \geq 0$$

*Proof.* We have

$$
\begin{aligned}
\mathbb{E}\left\|\bar{x}_{k+1}\right\|^2 &= \mathbb{E}\left\|\bar{x}_k - h\nabla f(\bar{x}_k) + \sqrt{2h}\xi_{k+1}\right\|^2 \\
&\overset{(i)}{=} \mathbb{E}\left\|\bar{x}_k\right\|^2 + h^2\mathbb{E}\left\|\nabla f(\bar{x}_k)\right\|^2 + 2hd - 2h\mathbb{E}\langle\bar{x}_k, \nabla f(\bar{x}_k)\rangle \\
&= \mathbb{E}\left\|\bar{x}_k\right\|^2 + h^2\mathbb{E}\left\|\nabla f(\bar{x}_k) - \nabla f(0)\right\|^2 + 2hd - 2h\mathbb{E}\langle\bar{x}_k, \nabla f(\bar{x}_k)\rangle \\
&\overset{(ii)}{\leq} \mathbb{E}\left\|\bar{x}_k\right\|^2 + h^2 L^2\mathbb{E}\left\|\bar{x}_k\right\|^2 + 2hd - 2h\mathbb{E}\langle\bar{x}_k, \nabla f(\bar{x}_k)\rangle \\
&\overset{(iii)}{\leq} \mathbb{E}\left\|\bar{x}_k\right\|^2 + h^2 L^2\mathbb{E}\left\|\bar{x}_k\right\|^2 + 2hd - 2mh\mathbb{E}\left\|\bar{x}_k\right\|^2 \\
&\overset{(iv)}{\leq} \left(1 - \frac{7}{4}mh\right)\mathbb{E}\left\|\bar{x}_k\right\|^2 + 2hd
\end{aligned}
$$

where $(i)$ is due to the independence between $\xi_{k+1}$ and $\bar{x}_k$, $(ii)$ is due to Assumption 1, $(iii)$ is due to the property of $m$-strongly-convex functions, $\langle\nabla f(y) - \nabla f(x), y - x\rangle \geq m\left\|y - x\right\|^2 \; \forall x, y \in \mathbb{R}^d$, and $(iv)$ uses the assumption $h \leq \frac{1}{4\kappa L}$.

Unfolding the inequality, we obtain

$$\mathbb{E}\left\|\bar{x}_k\right\|^2 \leq (1 - \frac{7}{4}mh)^k\mathbb{E}\left\|\bar{x}_0\right\|^2 + 2hd\left(1 + \frac{7}{4}mh + \cdots + (\frac{7}{4}mh)^{k-1}\right) \leq \mathbb{E}\left\|x_0\right\|^2 + \frac{8d}{7m}$$

$\qquad\square$

