# OpenReview forum: "Sqrt(d) Dimension Dependence of Langevin Monte Carlo"
_ICLR.cc/2022/Conference — ICLR 2022 Poster_

### Official Review · Reviewer_WTWj · 2021-11-02

**Correctness:** 4
**Technical Novelty And Significance:** 3
**Empirical Novelty And Significance:** 3
**Recommendation:** 8
**Confidence:** 4

**Main Review:**

The first result of the paper extends the already significant analysis of [Li et al. (2019)](https://proceedings.neurips.cc/paper/2019/hash/7d265aa7147bd3913fb84c7963a209d1-Abstract.html) and, in particular, only requires the local errors to be bounded by $C + D\text{ }\mathbb{E}[||\boldsymbol{x}||^2]^\frac{1}{2}$ – which is in line with the classical mean-square analysis of SDEs. Since it is not necessary to establish global $L^2$ bounds of the numerical solution to apply Theorem 3.3, I believe this framework would be particularly helpful for more complicated SDEs and numerical schemes, which seem to becoming more prevalent in the literature.

The second result is also considerable as it establishes the first $\mathcal{O}(\sqrt{d}h)$ 2-Wasserstein bound for ULA. The addition smoothness requirement, namely linear growth of $||\nabla(\Delta f)||$ allows $f$ to grow like $x^p$ with $p\leq 4$ and is less restrictive than the usual Lipschitz assumption on $\nabla^2 f$. It is also immediately clear in the proof why the term $\nabla(\Delta f)$ is important (it comes from Itô's lemma).

The paper and its appendix are clearly written and the analysis looks mathematically correct. Although the results of the paper are heavily based on previous works, I believe they are novel, provide significant improvements and would be of great interest to the community. For example, I can envision plenty of future research that would use Theorem 3.3 to analyse new SDE-based sampling algorithms. For ULA specifically, the condition A.2. looks appealing and it would be interesting to see which sampling problems have $f$ satisfying it. Thus, I believe this is a strong theory-based paper and would recommend it for acceptance into ICLR.

I could not identify any major weaknesses of the paper, and only have the following minor points / typos to discuss:

### Minor points
* In the numerical examples, it would also be nice to see how $\mathbb{E}[||\bar{\boldsymbol{x}}_k - \boldsymbol{x}_T||_2^2]^\frac{1}{2}$ scales with $d$ and $h$. This would require simulating $\bar{\boldsymbol{x}}$ and $\boldsymbol{x}$ with the same Brownian paths, but would result in the paper demonstrating the complexity of both upper and lower bounds for $W_2\big(Law(\bar{\boldsymbol{x}}_k), \mu\big)$.

* I think it should be made clear that the $p_2 \leq 1.5$ barrier for "increment only" methods is not the case in general. For general SDEs, the barrier is $p_2\leq 1$ (I think this was first shown in [Clark and Cameron (1980)](https://link.springer.com/content/pdf/10.1007/BFb0004007.pdf)). If one allows methods to also use the integral of Brownian motion against time, then the barrier becomes $p_2 \leq 2$ for additive noise SDEs ([Rößler (2010)](https://epubs.siam.org/doi/abs/10.1137/09076636X) gives very efficient stochastic Runge-Kutta methods for such SDEs). However the underdamped Langevin SDE considered by [Shen & Lee (2019)](https://papers.nips.cc/paper/8483-the-randomized-midpoint-method-for-log-concave-sampling) has special structure that allows for $p_2 > 2$ (for example, the schemes in [Sanz-Serna et al. (2021)](https://arxiv.org/abs/2104.12384) and [Foster et al. (2021)](https://arxiv.org/abs/2101.03446) have $p_2 = 2.5$ and $p_2 = 3.5$ respectively). In general, I'm not sure if randomization on its own leads to higher order convergence rates. For the randomized midpoint method, I believe it is the use of stochastic integrals that gives $p_2 = 2$, whilst the randomization allows for $p_1 = 3$ with just Lipchitz regularity of $\nabla f$. (pages 6 and 8)

* I feel that $\boldsymbol{z}$ should be written with a time dependence, i.e. $\boldsymbol{z}_t := (\boldsymbol{x}_t - \boldsymbol{x}) - (\boldsymbol{y}_t - \boldsymbol{y})$. It should also be clear how $\boldsymbol{z}$ is defined in equation (19) as $\boldsymbol{y}$ isn't used there. (pages 4 and 13)

* Perhaps say "it follows by Grönwall's inequality" instead of "it follows that" on page 17?

* In the proof of Lemma D.2, I think it would be worth mentioning that the Laplacian $\Delta$ and gradient $\nabla$ operators commute for $\mathcal{C}^3$ functions since Itô's lemma would ordinarily have $\Delta(\nabla f)$ in the correction term. (page 20)

### Typos
* "corresponds to an Euler-Maruyama discretization" instead of “corresponds to Euler-Maruyama discretization" (page 1)
* "distances/divergences" instead of "distances/diverges" (page 1)
* $\frac{h^2}{2} + o(h^2)$ could use be in larger brackets (page 5)
* "$W_2(Law(\boldsymbol{x}_0), \mu)$ instead of "$W_2(Law(\boldsymbol{x}_0\mu)$"  (page 5)
* Some equations are missing full stops or commas (pages 5, 6, 15,16 and 18). On page 15, the "$C \leq\ldots$" line is not aligned with the "$C =\ldots$" line above
* $\max$ and $\min$ could use larger curly brackets (pages 7,14 and 15)
* $\widetilde{\Omega}$ appears in Theorem 4.3 on page 8 but $\widetilde{\Theta}$ is used in its proof on page 16. In any case, I think it should be explained what this notation means in Section 2.
* "$W_2$" instead of "W2" (page 8)
* "error. Both" instead of "error.Both" (page 9)
* "SDE-based" instead of "SDE-basd" (page 9)
* "$L_t \leq L_0\exp(-2mt)$rea" (page 17)
* "by the conditioning version of" instead of "by conditioning version of" (page 17)
* $\mathbb{E}||\boldsymbol{x}||^2$ instead of $||\boldsymbol{x}||^2$ (page 20)
* $1-\frac{7}{4}mh$ could use large brackets (page 21)

**Summary Of The Paper:**

The paper is concerned with the non-asymptotic analysis of SDE-based sampling algorithms and has two main contributions. Firstly, it improves upon the general framework of [Li et al. (2019)](https://proceedings.neurips.cc/paper/2019/hash/7d265aa7147bd3913fb84c7963a209d1-Abstract.html) and, in particular, does not require uniform boundedness assumptions for the local strong/weak errors. This means their first result, Theorem 3.3, is easier to apply to numerical schemes for contractive SDEs and can already be seen to unify previous works (such as the analysis of KLMC in [Dalalyan et al. (2020)](https://projecteuclid.org/journals/bernoulli/volume-26/issue-3/On-sampling-from-a-log-concave-density-using-kinetic-Langevin/10.3150/19-BEJ1178.short)). Secondly, the authors consider the Unadjusted Langevin Algorithm (ULA) and derive a 2-Wasserstein bound of $\mathcal{O}(\sqrt{d} h)$, which has optimal dependence on both the dimension $d$ and step size $h$. To achieve this, they impose an additional smoothness condition on $f$ that is less restrictive than having the Hessian $\nabla^2 f$ Lipschitz continuous. The authors support this theory with numerical examples, where they demonstrate that a lower bound on the 2-Wasserstein error scales as $\mathcal{O}(\sqrt{d})$ and $\mathcal{O}(h)$.

**Summary Of The Review:**

The two main theoretical contributions of the paper are novel, well presented and improve upon previous results. The general mean-square analysis framework is likely to be applicable to a wide variety of SDEs / numerical schemes, and thus would aid future research in SDE-based sampling algorithms. The new $2$-Wasserstein bound on ULA scales optimally with dimension and step size, whilst requiring a smoothness assumption on $f$ that is weaker than the standard Lipschitz Hessian condition. Overall, I believe this is a strong theory-based paper and would recommend it for acceptance into ICLR.

---

> ### Author Response · Authors · 2021-11-21
> **Response to Reviewer WTWj**
>
> We sincerely thank the reviewer for carefully reading the paper and providing so many valuable comments which greatly improve this work. The confirmation of the significance of our work is also deeply appreciated. Below is an itemized list of responses.
>
> > numerical verification on how $\mathbb{E}\|\bar{\boldsymbol{x}}_k - \boldsymbol{x}_T\|^2$ scales with $d$ and $h$
>
> Indeed this would be very helpful. On the other hand, as the expert pointed out, this calculation is computationally challenging as it requires simulating $\bar{\boldsymbol{x}}$ and $\boldsymbol{x}$ with the same Brownian paths, and in order to get sufficiently close to the exact solution $\boldsymbol{x}$, tiny stepsize is needed, in addition to the necessity of running a large number of i.i.d. realizations for the empirical average. We gave some try but then estimated that we might not be able to finish it, given ICLR timeline and our computational resources. Fortunately we have a theoretical upper bounds of this so that our LMC complexity upper bound still holds true, and we also have an example to show that its tightness.
>
> > (a collection of important comments on) order and order barrier under various setups
>
> We deeply appreciate the discussion. Here are our detailed responses:
> 1. Re: order barrier for "increment only" methods.
> If we did not misunderstand, Clark and Cameron (1980) has the same result as what we stated, because what it showed is a global order barrier of $\leq 1$, whereas what we quoted is a barrier of local strong order $p_2 \leq 1.5$, and mean-square analysis framework stats that the global order is $p_2-0.5$ (if $p_1\geq p_2+0.5$). Nevertheless, 1980 is earlier than Rüemelin (1982) which is what we cited, so we definitely should include that citation. It is very much appreciated and now added.
> 2. Re: higher order, and Rößler (2010).
> Yes, Andreas Rößler had a series of classical papers from which we learned a lot, and indeed with multiple stochastic integrals, orders can be improved; the resulting methods in general can be awkward but the expert reviewer is absolutely right that additive noise helps a lot. We added this citation to our previous statement in the paper that methods involving multiple stochastic integrals can yield a larger $p_2$.
> 3. underdamped Langevin has special structure.
> Thanks for the insightful comment and the very interesting references! We completely agree.
> 4. effects of randomization v.s. special structure in underdamped Langevin.
> Thanks again for the insight! We are not yet sure what is the limit of the capability of randomization, but this is a very important question. We actually have been researching, but no quantitative results yet. We revised the statement in the paper to reflect this fact.
> 5. orders of randomized midpoint for underdamped Langevin.
> Our preliminary calculation actually suggests $p_1=4$ but since the expert reviewer mentioned $p_1=3$, we need to check again. Fortunately, this is parallel to this paper so we will do it carefully later.
>
> > additional minor comments and typos
>
> The other 3 minor comments and 14 typos are also deeply appreciated. They are now addressed/corrected in the updated version.

---

### Official Review · Reviewer_XyDg · 2021-11-02

**Correctness:** 3
**Technical Novelty And Significance:** 3
**Empirical Novelty And Significance:** 2
**Recommendation:** 6
**Confidence:** 4

**Main Review:**

I think this is a good piece of improvement but can benefit from reorganisation and clarification of the results.

The main result of this paper is the improved complexity bound, i.e., it is established that in order for LMC to attain $\epsilon$ error, the number of iterations scale $\mathcal{O}(\sqrt{d})$, not $\mathcal{O}(d)$ as proved in previous works. This closes the gap between second-order methods like underdamped Langevin Monte Carlo which are claimed to be better in this sense -- this paper effectively shows that this claim is not correct (at least in terms of dimension).

Here are my suggestions.

1) I think the paper can benefit from a clean organization, i.e., stating the main results in Section 2, instead of leaving them to Section 4. It must be also clarified in this sense why the mean-square analysis is precisely useful for $W_2$ (as the mean square error upper bounds $W_2$). I think Section 2 must be devoted to main results and especially the non-standard assumption made in this paper to attain this result.

2) The authors should clarify in a remark how precisely A2 helps to get the result. In other words, which part of the proof fails if this assumption is dropped? Is it possible to use the "mean-square analysis" without this assumption and get $\mathcal{O}(d)$ dependence anyway?

3) It is nice to have the empirical demonstrations and seeing the scaling $\mathcal{O}(\sqrt{d})$ in practice. However, does A2 hold for these examples? It has to be shown that the analysis is valid for these simple examples. Another alternative, if this is not possible, is to provide a more complicated example where the scaling still can be observed.

**Summary Of The Paper:**

This paper derives an error bound for the LMC that's improved in dimension compared to earlier bounds, at the cost of an extra assumption.

**Summary Of The Review:**

This is a good paper that can benefit from a clarification and improvements.

---

> ### Author Response · Authors · 2021-11-21
> **Response to Reviewer XyDg**
>
> We deeply thank the reviewer for valuable comments that help improve this paper. Here is an itemized list of our responses:
>
> 1. Thanks to the reviewer's suggestion, we added pointers in Section 2 so that main results are described early on.
> 2. Thanks to the reviewer's suggestion, we added an additional remark on the 3rd-order growth assumption, to explain how it is used in the proof and why it is important. This assumption, together with Ito's lemma, helps establish an order $p_1=2$ of local weak error for LMC. Without this assumption, the current analysis tools that we know can only help show an $\mathcal{O}(d)$ dependence for LMC (see the remark below Theorem 4.2 and also Example 1 in [Li et al., 2019]).
> 3. Both examples in our experiments satisfy A2 as mentioned in the paper (right after the expressions of $f_1$ and $f_2$ in Sec.5). We apologize to have buried that in the text.

---

### Official Review · Reviewer_uCPh · 2021-11-03

**Correctness:** 4
**Technical Novelty And Significance:** 3
**Empirical Novelty And Significance:** Not applicable
**Recommendation:** 8
**Confidence:** 4

**Main Review:**

The main strength of the paper is to get sqrt(d) dependence on a dimension with not very restrictive assumptions.
In my opinion, the linear growth condition of the third derivative could be discussed in more detail, to improve the presentation of the results.


**Summary Of The Paper:**

The authors provide new non-asymptotic bounds for the Langevin Mont Carlo algorithm in the strongly convex settings with additional growth conditions for third-order derivatives. The additional assumption allows to get sqrt(d) dependence on the dimension, which is optimal since it meets the exact value for the Gaussian case.

**Summary Of The Review:**

I found the paper interesting and I recommend accepting it. I think that a deeper  discussion on assumptions could help readers to better understand main results.

---

> ### Author Response · Authors · 2021-11-21
> **Response to Reviewer uCPh**
>
> We sincerely thank the reviewer's comment. The positive expert report is very much appreciated.
>
> Regarding the 3rd-order growth condition, in addition to the existing three remarks, we added one more. We hope these remarks could further improve the readability of the paper, and thank you again for the helpful note.

---

### Official Review · Reviewer_LMTg · 2021-11-04

**Correctness:** 4
**Technical Novelty And Significance:** 3
**Empirical Novelty And Significance:** 3
**Recommendation:** 6
**Confidence:** 3

**Main Review:**

I thank the authors for their article, which addresses the important topic of analytically understanding the convergence of the unadjusted Langevin Monte Carlo (LMC) algorithm.


Originality: The authors improve upon existing work to obtain a 2-Wasserstein mixing time bound of O(d^0.5/\epilson) for LMC, and show this is optimal. This is an original and significant contribution.

Clarity: The article could be better written. For example, incorporating some of the remarks into the main text may read better.

Other than presentation, I do not have any other direct comments about the paper. Note: I have looked through the proofs, but not in great detail.

**Summary Of The Paper:**

The manuscript considers the unadjusted Langevin Monte Carlo (LMC) algorithm, and performs non-asymptotic analysis of its convergence with respect to the 2 Wasserstein distance. The main contribution is a mixing time bound of O(d^0.5/\epilson), which improves upon the existing O(d/\epsilon) bound of Durmus and Moulines. The O(d^0.5/\epilson) is also shown to be optimal for the classes of probability distributions considered.

**Summary Of The Review:**

The manuscript establishes new improved non-asymptotic O(d^0.5/\epilson) convergence rates for the unadjusted Langevin Monte Carlo (LMC) algorithm. This is a significant contribution and will be of interest to the ICLR community.

---

> ### Author Response · Authors · 2021-11-21
> **Response to Reviewer LMTg**
>
> We greatly appreciate the reviewer's comments. Glad to know that the originality and significance of our work are recognized.
>
> Per the reviewer's suggestion, we incorporated a few remarks into the main text for improved readability.

---

### Decision · Program_Chairs · 2022-01-20

**Decision:**

Accept (Poster)

**Comment:**

This paper provides a near-optimal analysis of the unadjusted Langevin Monte Carlo (LMC) algorithm with respect to the W2 distance. The main statement is that the mixing time is ~ d^{1/2}/eps under standard assumptions. The authors also give a nearly matching lower bound under these assumptions. The reviewers agreed that this is an interesting contribution obtained via non-trivial techniques. The consensus recommendation is to accept the paper.